# PPP1R15A-mediated dephosphorylation of eIF2α is unaffected by Sephin1 or Guanabenz

Ana Crespillo-Casado[1]*, Joseph E Chambers[1], Peter M Fischer[2,3], Stefan J Marciniak[1], David Ron[1]*

[1]Cambridge Institute for Medical Research, University of Cambridge, Cambridge, United Kingdom; [2]Division of Biomolecular Science and Medicinal Chemistry, School of Pharmacy, University of Nottingham, Nottingham, United Kingdom; [3]Centre for Biomolecular Sciences, University of Nottingham, Nottingham, United Kingdom

**Abstract** Dephosphorylation of translation initiation factor 2 (eIF2α) terminates signalling in the mammalian integrated stress response (ISR) and has emerged as a promising target for modifying the course of protein misfolding diseases. The [(o-chlorobenzylidene)amino]guanidines (Guanabenz and Sephin1) have been proposed to exert protective effects against misfolding by interfering with eIF2α-P dephosphorylation through selective disruption of a PP1-PPP1R15A holophosphatase complex. Surprisingly, they proved inert in vitro affecting neither stability of the PP1-PPP1R15A complex nor substrate-specific dephosphorylation. Furthermore, eIF2α-P dephosphorylation, assessed by a kinase shut-off experiment, progressed normally in Sephin1-treated cells. Consistent with its role in defending proteostasis, Sephin1 attenuated the IRE1 branch of the endoplasmic reticulum unfolded protein response. However, repression was noted in both wildtype and *Ppp1r15a* deleted cells and in cells rendered ISR-deficient by CRISPR editing of the *Eif2s1* locus to encode a non-phosphorylatable eIF2α (eIF2α$^{S51A}$). These findings challenge the view that [(o-chlorobenzylidene)amino]guanidines restore proteostasis by interfering with eIF2α-P dephosphorylation.

*For correspondence: ac880@cam.ac.uk (AC-C); dr360@medschl.cam.ac.uk (DR)

## Introduction

Protein folding homeostasis (proteostasis) is achieved by balancing the rate of production, folding and protein degradation. Proteostasis is strongly influenced by the phosphorylation state of serine 51 of the α subunit of eukaryotic translation initiation factor 2 (eIF2α) (*Sonenberg and Hinnebusch, 2009*). Diverse stress conditions activate kinases that phosphorylate eIF2α, resulting in attenuated rates of translation initiation of most mRNAs and increasing translation of a small group of mRNAs with special 5′ untranslated regions (*Hinnebusch, 2014*). The latter encode potent transcription factors such as ATF4 that couple eIF2α phosphorylation to the Integrated Stress Response (ISR) (*Harding et al., 2003*), a transcriptional and translational program that adapts cells to stress and participates in diverse biological processes such as memory, immunity and metabolism (*Baird and Wek, 2012*).

Signalling in the ISR is terminated by eIF2α-P dephosphorylation. This requires the presence of a regulatory subunit, PPP1R15, to direct the catalytic, PP1 subunit, to its specific substrate. Two mammalian genes encode PPP1R15 regulatory subunits. *Ppp1r15b* (or CReP) encodes a constitutively expressed regulatory subunit (*Jousse et al., 2003*), whereas *Ppp1r15a* (or GADD34) encodes an ISR inducible regulatory subunit that contributes to a negative feed-back loop operative in the ISR (*Brush et al., 2003*; *Ma and Hendershot, 2003*; *Novoa et al., 2001*, *2003*). A minimal rate of

**eLife digest** Most drugs work by tweaking the way that cells are regulated. Adding or removing a phosphate group from proteins regulates many cellular decisions. There are known drugs that bind to and inhibit the enzymes that add phosphate to proteins, thereby controlling various aspects of cell behaviour. However, drug developers have been far less successful in finding drugs that inhibit phosphatases, the enzymes that remove phosphate from proteins.

Genetically modified mice can be used as 'models' to investigate human diseases. In 2015 a drug called Sephin1 was reported to suppress neurodegeneration in a group of these mice by inhibiting a particular phosphatase. The phosphatase is made of three component proteins that come together to create the active enzyme. Sephin1 was reported to disrupt the association between two of these three components. This discovery was met with excitement; both for its potential therapeutic implications in humans and as an important "first" in pharmacology.

To understand how Sephin1 and a related drug, Guanabenz, work at the molecular level, Crespillo-Casado et al. reconstructed in a test tube the phosphatase that Sephin1 and Guanabenz were reported to inhibit. To examine the effects the drugs have on the phosphatase, Crespillo-Casado et al. developed assays to measure the association between the components that make up the phosphatase. Further assays measured the removal of phosphate from the phosphatase's target, a protein called eIF2α.

The results of the assays show that Sephin1 did not affect the coming together of the components that make up the active phosphatase. The drug also did not inhibit the removal of phosphate from eIF2α in the test tube. To extend these findings Crespillo-Casado et al. exposed cells to Sephin1 and observed features that are consistent with the drug's reported ability to supress neurodegeneration. However, these features were also observed both in cells lacking the phosphatase that Sephin1 was reported to inhibit and in cells in which eIF2α never acquired a phosphate in the first place.

The findings presented by Crespillo-Casado et al. do not challenge Sephin1's role in supressing neurodegeneration, but do question its ability to do so by inhibiting the phosphatase that dephosphorylates eIF2α. This knowledge will be useful to drug developers and those interested in molecular mechanisms of drug action. For those researchers who are interested in Sephin1, further work is needed to discover alternative molecular mechanisms by which it suppresses neurodegeneration. And for those researchers who are interested in eIF2α dephosphorylation, there is a need to look further for inhibitors of this process, as Sephin1 is unlikely to serve in that role.

eIF2α-P dephosphorylation is an essential cellular function, as reflected in the severe phenotypes of *Ppp1r15b* mutation or deletion and in the very early lethality of compound *Ppp1r15a;b* deficient mice (*Abdulkarim et al., 2015*; *Harding et al., 2009*).

Interestingly, whilst deletion of the inducible *Ppp1r15a* gene results in sluggish recovery of protein synthesis during the waning phase of stress (*Kojima et al., 2003*; *Novoa et al., 2003*), mice lacking any PPP1R15A-directed eIF2α-P dephosphorylation (homozygous *Ppp1r15a^tm1Dron*) are superficially indistinguishable from wildtype. Moreover, when challenged with tunicamycin, which causes unfolded protein stress in the endoplasmic reticulum by inhibiting N-linked glycosylation, homozygous *Ppp1r15a^tm1Dron* mice and cultured cells derived from them are relatively resistant to the toxin's lethal effects (*Marciniak et al., 2004*; *Reid et al., 2016*). This feature is plausibly attributed to sustained activity of the ISR in the *Ppp1r15a* mutant mice, which favours proteostasis by limiting the production of unfolded proteins under stress conditions (*Boyce et al., 2005*; *Han et al., 2013*).

The proteostasis-promoting features of interfering with PPP1R15A-mediated eIF2α-P dephosphorylation are also played out in the context of certain disease models associated with protein misfolding and proteotoxicity. Both the neuropathic phenotype associated with Schwann cell expression of a mutant misfolding-prone myelin constituent, P0^S63Δ, and a mutant superoxide dismutase expressed in motor neurones are ameliorated by a concomitant dephosphorylation-defective *Ppp1r15a^tm1Dron* mutation (*D'Antonio et al., 2013*; *Wang et al., 2014*), and similar amelioration of

inflammatory-mediated central nervous system demyelination is observed in the *Ppp1r15a[tm1Dron]* mice (*Lin et al., 2008*).

These features have led to an interest in the therapeutic potential of targeting PPP1R15-mediated eIF2α-P dephosphorylation with small molecule inhibitors. Early work led to discovery of salubrinal, a small molecule that increases levels of eIF2α-P and retards its dephosphorylation. However, salubrinal is only known to work in vivo and its mechanism of action remains unclear (*Boyce et al., 2005*). Limitations of in vitro assays for substrate-specific PPP1R15-mediated eIF2α-P dephosphorylation (see below) have all but precluded a biochemical approach to the problem, but a cell based search for proteostasis regulators suggested that the α2 adrenergic blocker Guanabenz, [(*o,o*-dichlorobenzylidene)amino]guanidine, might exert its beneficial effects on proteotoxicity by interfering with eIF2α-P dephosphorylation (*Tsaytler et al., 2011*). This theme was extended further by the discovery of Sephin1, [(*o*-chlorobenzylidene)amino]guanidine, that had lost its α2 adrenergic blocking activity but retained its proteostasis promoting properties (*Das et al., 2015*). Importantly, biochemical characterization of Guanabenz and Sephin1 suggested that both disrupt the complex between PPP1R15A and PP1, providing strong support for a mechanism of action that involves interfering with PPP1R15A-mediated eIF2α-P dephosphorylation (*Das et al., 2015*), *Figure 1C* therein).

Genetic analysis reveals that a regulatory PPP1R15 subunit is essential for eIF2α-P dephosphorylation in vivo, and over-expression of either PPP1R15A or PPP1R15B or merely their conserved C-terminal portion, is sufficient to deregulate eIF2α-P dephosphorylation in vivo and inhibit the ISR (*Brown et al., 1997*; *Brush et al., 2003*; *Jousse et al., 2003*; *Novoa et al., 2001*). Though PPP1R15 regulatory subunits stably bind the catalytic subunit (PP1), the resulting binary complex is devoid of specificity towards eIF2α-P. However, G-actin joins the PPP1R15-PP1 binary complex as an ancillary subunit to form a ternary complex endowed with substrate-specific eIF2α-P directed phosphatase activity, both in cells (*Chambers et al., 2015*) and when constituted with pure components in vitro (*Chen et al., 2015*).

To explore in detail the mechanism of action of the [(*o*-chlorobenzylidene)amino]guanidines we reconstructed PPP1R15A-PP1-G-actin-mediated eIF2α-P dephosphorylation in vitro with pure components. Using mutants that interfere with complex formation and function we established the correlation of enzymatic activity with the kinetic parameters of complex formation to develop an assay responsive to the stability of the core PPP1R15-PP1 binary interaction; the proposed target of Guanabenz and Sephin1. The results of our inquiry, reported on below, question the role of destabilization of the eIF2α-P directed holophosphatase in the proteostatic effects of these compounds.

## Results

### In vitro assay for selective eIF2α-P dephosphorylation sensitive to the stability of the PPP1R15A-PP1 complex

PPP1R15A/GADD34 is a protein of >600 residues, but only the C-terminal 70 residues are necessary for substrate-specific dephosphorylation of eIF2α-P (*Figure 1A*). This active fragment is also the most conserved segment of the protein; both between homologues of PPP1R15A and with the paralogous PPP1R15B (*Figure 1B*). This region of the protein is natively unfolded (*Yu et al., 2004*), attaining its structure upon binding the PP1 catalytic and the G-actin ancillary subunits (*Chen et al., 2015*; *Choy et al., 2015*).

For structural studies we found it convenient to co-express PPP1R15 and PP1 in bacteria (*Chen et al., 2015*). However, the fixed 1:1 stoichiometry of the two subunits imparted by co-expression is unsuited to a detailed examination of the bimolecular affinities involved in complex formation or to the design of an assay sensitive to the stability of PPP1R15A-PP1 complex. To circumvent this limitation, we incorporated a highly soluble maltose-binding protein (MBP) moiety C-terminal to the natively-unstructured human PP1R15A active fragment. When expressed in *E. coli* as a fusion protein with a cleavable N-terminal glutathione S-transferase (GST) tag, GST-PPP1R15A-MBP remained soluble and when added as a purified protein in vitro (after cleavage of the GST), imparted eIF2α-P dephosphorylation activity to reactions containing purified PP1 and G-actin (*Figure 1C*, left panel). Moreover, the solubilizing MBP tag enabled recovery not only of a human PPP1R15A active fragment (residues 533–624) but also a much larger N-terminally extended fragment (residues 325–636). The minimal active fragment and the much longer N-terminally-extended PPP1R15A had similar

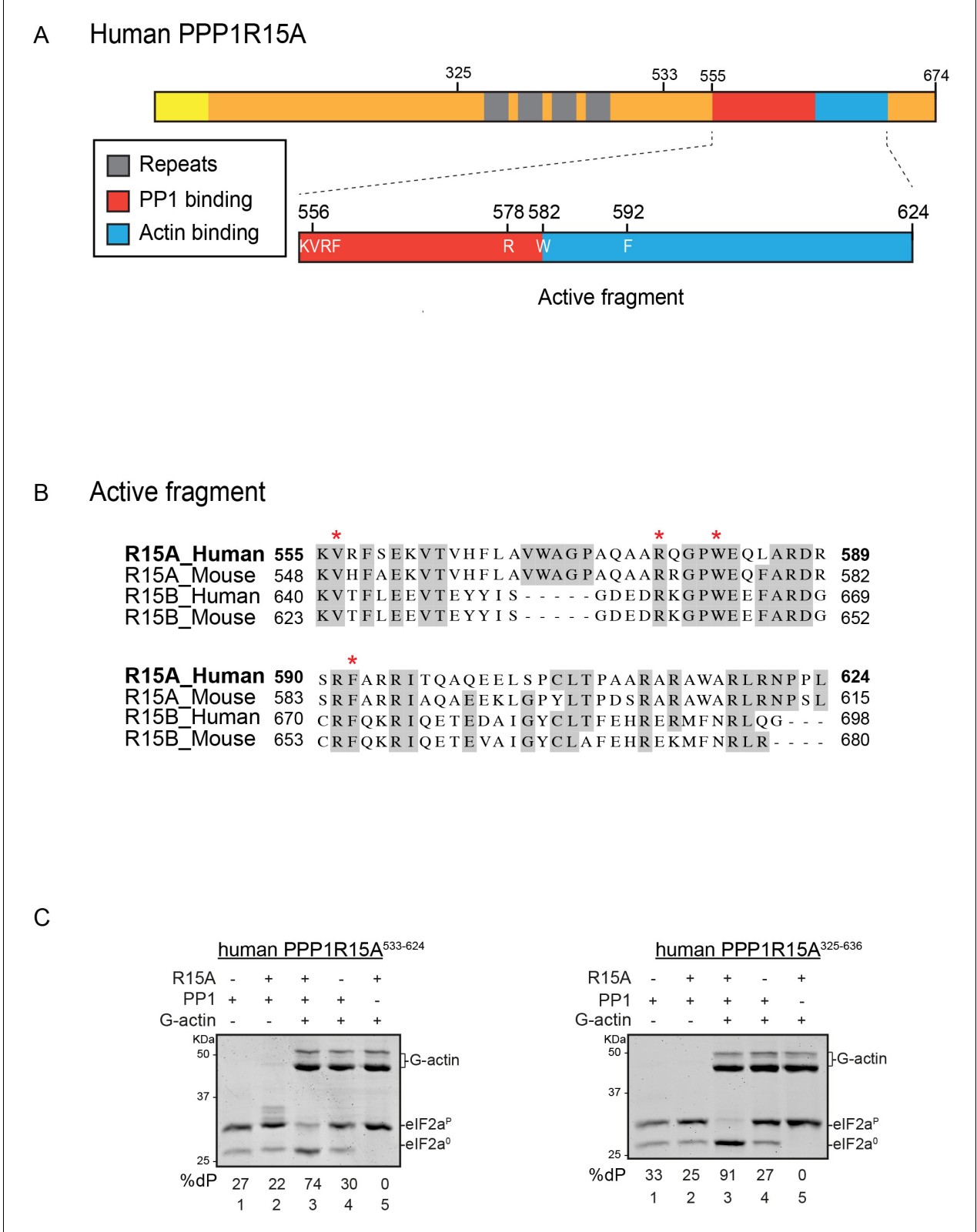

**Figure 1.** A tripartite assay for human PPP1R15A-dependent eIF2α-P dephosphorylation. (**A**) Cartoon representation of human PPP1R15A (GADD34). The minimal C-terminal peptide required for eIF2α-P dephosphorylation is outlined ('active fragment') and key residues in the PP1 and G-actin binding regions are annotated. (**B**) Alignment of C-terminal active fragments of mammalian PPP1R15A and PPP1R15B (CREP) using ClustalX. Grey highlighted residues represent conserved or highly similar residues. Red asterisks highlight key residues that are analysed in further detail. (**C**) Image of Coomassie-

*Figure 1 continued on next page*

*Figure 1 continued*

stained PhosTag-SDS-PAGE on which phosphorylated (eIF2α$^P$) and non-phosphorylated (eIF2α$^0$) forms of eIF2α from 20 min dephosphorylation reactions were resolved. The composition of the reaction, PPP1R15A$^{533-624}$ [80 nM], PPP1R15A$^{325-636}$ [400 nM], PP1c [12.5 nM] and G-actin [750 nM] are noted above and the fraction of dephosphorylated eIF2α is noted underneath each lane (%dP). The migration of molecular weight markers and G-actin are noted (the signal from the PPP1R15A and PP1c is undetectable on these gels).

activity in this assay (*Figure 1C*). Nonetheless the ability to purify a soluble, N-terminally extended PPP1R15A regulatory subunit expanded the possibilities to study more physiological models of PPP1R15A-mediated eIF2α-P dephosphorylation (a point we shall return to below).

We quantified the dependence of eIF2α-P dephosphorylation rates on both the concentration of the regulatory human PPP1R15A subunit (EC$_{50}$ = 7 nM) and on the ancillary G-actin subunit (EC$_{50}$ = 13 nM) (*Figure 2*). The latter values agreed with our previous measurements of G-actin's stimulation of enzymatic activity (in an assay using the murine PPP1R15A) (*Chen et al., 2015*), whereas the EC$_{50}$ of human PPP1R15A was within an order of magnitude of the affinity of human PPP1R15A for PP1, as measured by isothermal titration calorimetry (*Choy et al., 2015*) (see below).

Previous studies have identified mutations in PPP1R15 that abolish substrate-specific dephosphorylation in vitro and block PPP1R15's ability to repress the ISR, when expressed in vivo. The human PPP1R15A$^{V556E}$ mutation alters a key residue, part of the RVxF motif involved in binding of diverse regulatory subunits to PP1 (*Egloff et al., 1997*); its presence abolished all PPP1R15A-mediated eIF2α-P dephosphorylation (*Figure 3A*). Two previously-identified mutations in the C-terminal extension of PPP1R15A - the portion that interacts with the G-actin ancillary subunit (human PPP1R15A$^{W582A}$ and PPP1R15A$^{F592A}$) (*Chen et al., 2015*) - also abolished all PPP1R15A-mediated eIF2α-P dephosphorylation (*Figure 3B and C*). These findings establish the dependence of the tripartite assay described above on features known to be important for PPP1R15 function.

A fourth mutation tested affects a residue whose counterpart in human PPP1R15B$^{R658C}$ results in a syndromatic form of diabetes mellitus. Consistent with the destabilizing effect of this mutation on PP1 binding (*Abdulkarim et al., 2015*), its presence in human PPP1R15A$^{R578A}$ resulted in a ~4 fold increase in EC$_{50}$ for eIF2α-P dephosphorylation (*Figure 4A*). The mutation also affected the maximal stimulation afforded by human PPP1R15A$^{R578A}$, as even at saturating concentrations of regulatory subunit, eIF2α-P dephosphorylation reactions assembled with the mutant were three times slower than those assembled with the wildtype (*Figure 4B*). The human PPP1R15A$^{R578A}$ mutation does not appear to have a major effect on the stability of the G-actin containing ternary complex, as the EC$_{50}$ for G-actin (20 nM) was relatively unaffected (*Figure 4C*).

## Bio-Layer Interferometry measurement of the affinity of the holophosphatase components for each other

The features of the human PPP1R15A$^{R578A}$ mutant noted above suggest that the tripartite assay is sensitive not only to the affinity of the three components for one another but also to subtle structural features of the holophosphatase. To explore this issue further, we used Bio-Layer Interferometry (BLI) (*Abdiche et al., 2008*) to measure directly the affinity of the three components of the holophosphatase for each other. PPP1R15A$^{533-624}$ was biotinylated on a single lysine residue of an AviTag (*Fairhead and Howarth, 2015*) added between the cleavable GST tag and PPP1R15A peptide (*Figure 5A*) and the biotinylated protein was immobilized on a streptavidin-derivatized BLI biosensor tip. The biotinylated PPP1R15A$^{533-624}$ ligand showed a robust 1:1 bimolecular interaction, with pure PP1, yielding a $k_{off}$ = 0.21 ± 0.01 min$^{-1}$ and a $K_d$ = 20 ± 0.61 nM (*Figure 5B*). The higher affinity of PPP1R15 for PP1 observed here, compared to isothermal titration calorimetry (ITC) measurements of *Choy et al. (2015)* ($K_d$ = 62 ± 14 nM) might reflect the contribution of contacts made by residues C-terminal to PPP1R15A$^{L567}$, which are present in the construct used here, but absent from the one used in the ITC measurements (*Choy et al., 2015*). Cooperativity provided by G-actin (present in the enzymatic assay, but absent from the BLI experiment) and steric hindrance from probe components might have contributed to the 3–5 fold lower value of the PPP1R15A EC$_{50}$ for eIF2α-P dephosphorylation in the enzymatic assay (7 nM, *Figure 2B*) compared to the $K_d$ observed by BLI.

To gauge the affinity of the ancillary G-actin subunit for the complex, we first assembled a binary complex between the biotinylated PPP1R15A$^{533-624}$ ligand (described above) and a saturating

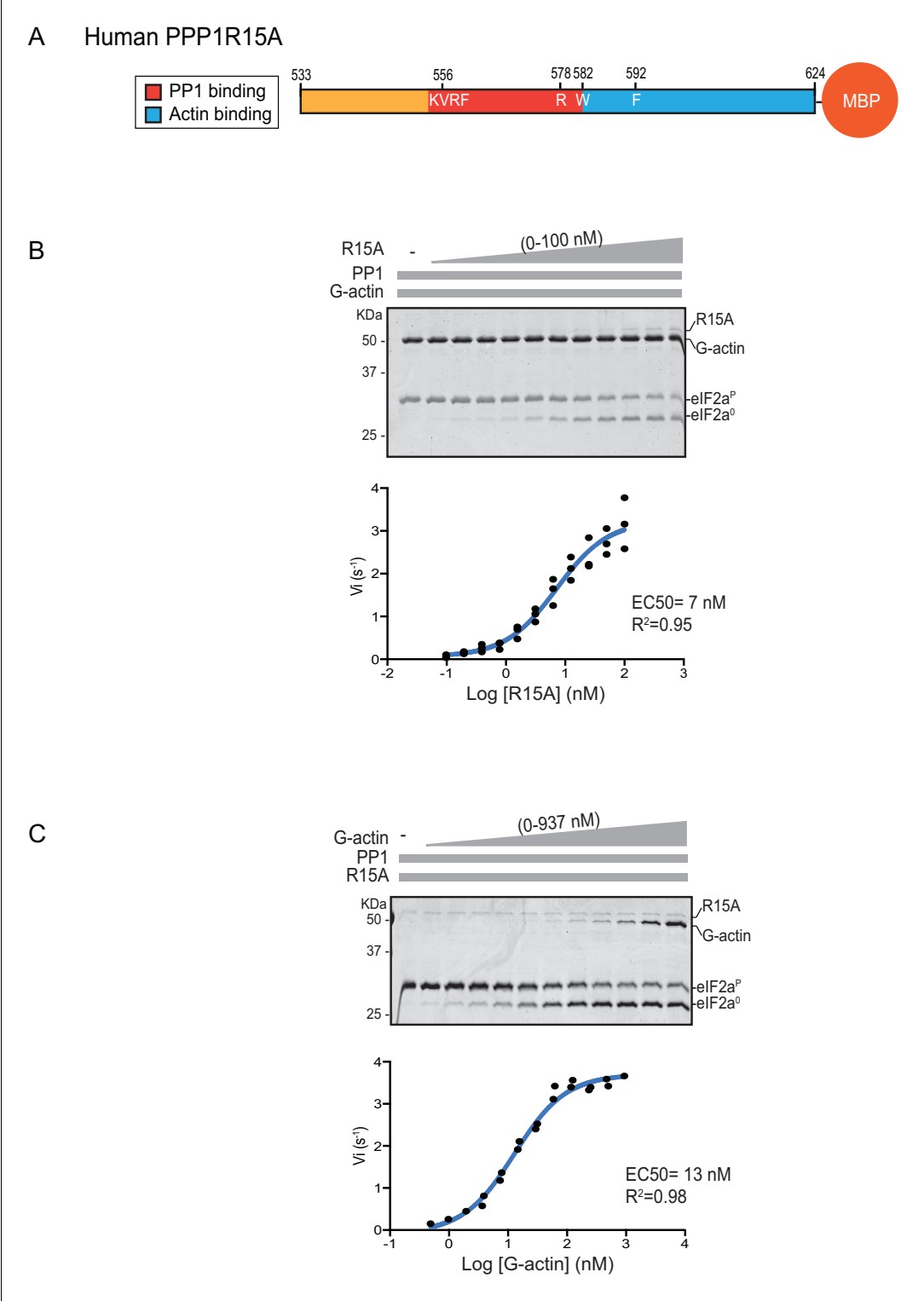

**Figure 2.** eIF2α-P dephosphorylation kinetics as a function of human PPP1R15A[533-624] and G-actin concentration. (A) Schema of the human PPP1R15A[533-624] construct used. The C-terminal Maltose Binding Protein (MBP) component, which stabilizes the fusion protein, is noted. (B) **Upper panel.** Coomassie-stained PhosTag-SDS-PAGE tracking the dephosphorylation of eIF2α[P] to eIF2α[0] in 20 min dephosphorylation reactions constituted with eIF2α[P] [2 μM], PP1 [0.625 nM], G-actin [1.5 μM] and an escalating concentration of PPP1R15A[533-624]. Shown is a representative of three independent experiments performed. **Lower panel**: Semi-log[10] plot of the initial velocity of eIF2α[P] dephosphorylation as a function of PPP1R15A[533-624] concentration derived from three repeats (one shown above). The EC[50] for PPP1R15A[533-624] was calculated using the agonist fitting function on

*Figure 2 continued on next page*

Figure 2 continued

GraphPad Prism V7. (**C**) Upper panel. As in 'B' but dephosphorylation of eIF2$\alpha^P$ to eIF2$\alpha^0$ was carried out in the presence of a fixed concentration of PPP1R15A$^{533-624}$ [50 nM] and an escalating concentration of G-actin. Shown is a representative of two independent experiments performed. **Lower panel**: Semi-log$_{10}$ plot of initial velocity as a function of G-actin concentration derived from two repeats (one shown above).

concentration of PP1 and then measured the BLI signal induced by addition of G-actin. A robust association-dissociation signal was observed with purified G-actin ($k_{off}$ = 2.84 ± 0.11 min$^{-1}$ and $K_d$ = 151 ± 14.3 nM) (**Figure 5C**).

The human PPP1R15A$^{V556E}$ mutation, affecting the RVxF motif, abolished all measureable association with PP1, but had no effect on the kinetics of G-actin binding. Conversely, the human PPP1R15A$^{F592A}$ mutation markedly enfeebled G-actin binding but had no effect on PP1 binding (**Figure 6A and B**). Together, these observations confirm the ability of PPP1R15A to engage PP1 and G-actin independently, via the N- and C-terminal parts of its active portion. Despite their strong detrimental effects on enzymatic activity (**Figures 3C** and **4**), neither the PPP1R15A$^{R578A}$ nor the PPP1R15A$^{W582A}$ mutations had a major effect on the kinetics of PP1 or G-actin binding (**Figure 6**). Together, these observations suggest that the tripartite enzymatic assay is sensitive both to mutations that grossly interfere with complex stability (V556E and F592A) and to mutations that more subtly affect the structure of the complex (R578A and W582A).

## No measureable effect of [(*o*-chlorobenzylidene)amino]guanidines on PPP1R15A-containing holophosphatases in vitro

Das and colleagues previously reported that addition of 50 µM Sephin1 to tissue culture media disrupts the PPP1R15A-PP1 complex recovered from cells (**Das et al., 2015**). To determine if these observations correlate with an effect of Sephin1 on the complex formed in vitro between PPP1R15A$^{533-624}$ and PP1, we sourced Sephin1 and confirmed its purity and identity by reverse phase HPLC and mass spectrometry (**Figure 7A**). When added to the BLI assay at a concentration of 50 µM (before exposure to PP1), Sephin1 had no measureable effect on either the association or dissociation phase of the assay (**Figure 7B**).

A biotinylated N-terminally extended PPP1R15A$^{325-636}$, corresponding to the construct studied by Das and colleagues, proved unsuited as a ligand in the BLI experiment. To circumvent this problem we biotinylated PP1 and exploited it as a BLI ligand. Addition of either the minimal active fragment, human PPP1R15A$^{533-624}$, or the longer human PPP1R15A$^{325-636}$, gave rise to a robust BLI signal but addition of Sephin1 affected neither the association nor dissociation phase of the experiment (**Figure 7C and D**). The kinetics of the bimolecular PP1-PPP1R15A interaction were reproducibly different when one or the other was used as a ligand (summarized in **Figure 7E**). These may reflect different distorting effect of other elements of the BLI biosensor on the kinetics of dissociation when PP1 or PPP1R15A were used as ligands, and/or a contribution of the N-terminal repeats of PPP1R15A to its interactions with PP1, as suggested previously (**Brush and Shenolikar, 2008**). However, reproducible inertness in all three assays lends confidence to the conclusion that in this experimental system 50 µM Sephin1 does not directly interfere with assembly or stability of the PPP1R15A-PP1 complex.

Next we sought to examine the effect of Sephin1 on the in vitro dephosphorylation activity of a tripartite eIF2$\alpha$-P holophosphatase assembled with the N-terminally extended PPP1R15A$^{325-636}$ (corresponding to the construct studied by Das and colleagues). As Sephin1 is proposed to inhibit eIF2$\alpha$-P dephosphorylation by disrupting the binding of PPP1R15A to PP1, we sought to incorporate the PPP1R15A$^{325-636}$ component at or below its EC$_{50}$, thereby maximizing the prospects of detecting an inhibitory effect. Addition of purified PPP1R15A$^{325-636}$ to PP1 and G-actin accelerated eIF2$\alpha$-P dephosphorylation with an EC$_{50}$ of 5–10 nM (**Figure 8A**). However, Sephin1 had no effect on the dephosphorylation reaction (**Figure 8B**), whilst tautomycin readily inhibited the reaction (IC$_{50}$ = 2.4 nM) confirming the sensitivity of the assay to a known inhibitor (**Figure 8C**). These observations were also confirmed in an assay set up with the corresponding murine PPP1R15A$^{273-657}$ (**Figure 8—figure supplement 1A and B**). The related compound Guanabenz also proved inert, even when added to the enzymatic assay at the high concentration of 50 µM (**Figure 8D**). Salubrinal, added at 12 µM (higher concentrations led to conspicuous precipitation) had a mild but highly reproducible inhibitory

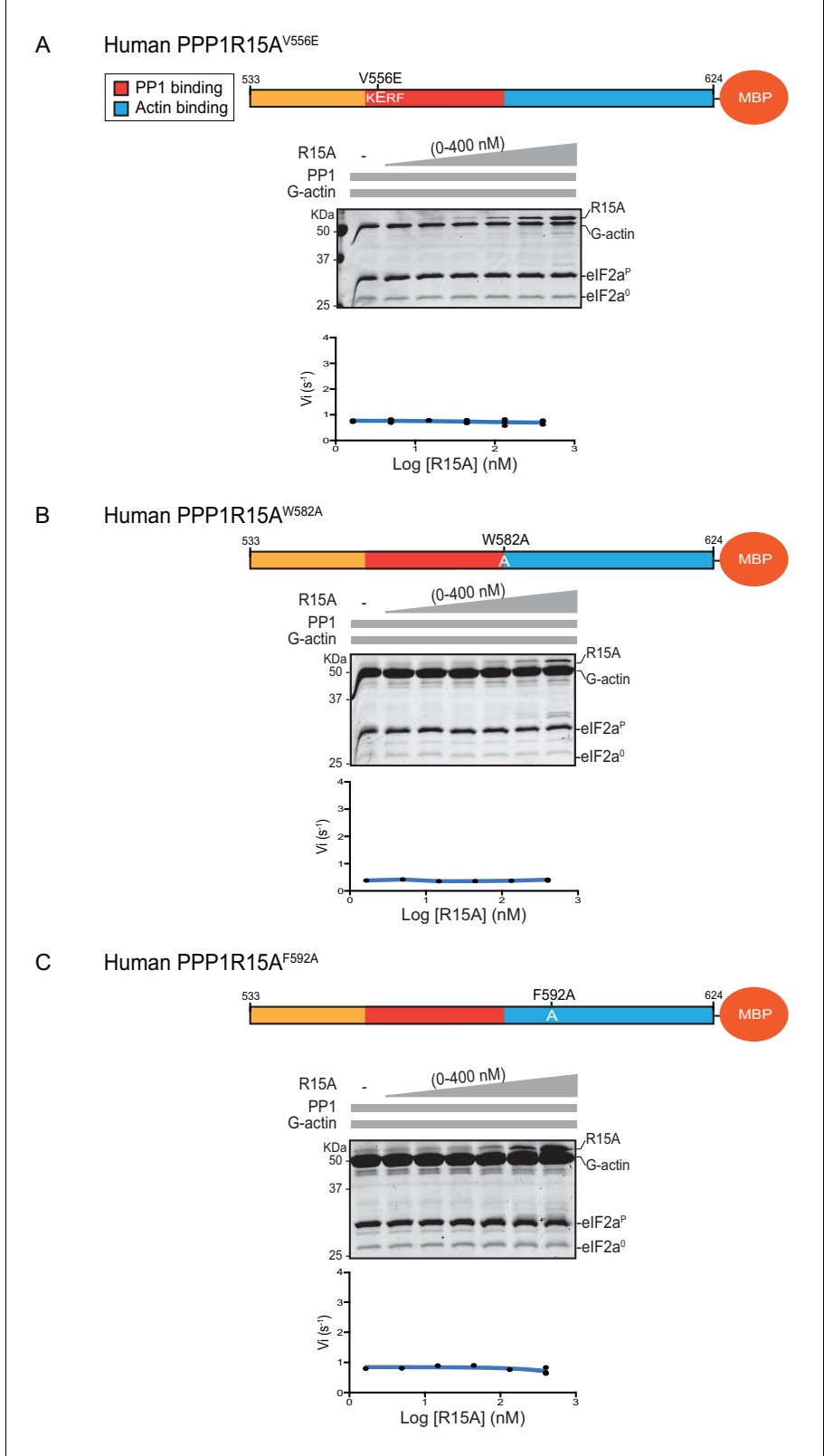

**Figure 3.** eIF2α-P dephosphorylation by ternary complexes constituted with human PPP1R15A[(533-624)V556E], PPP1R15A [(533-624)W582A] or PPP1R15A [(533-624)F592A]. (**A**) Coomassie-stained PhosTag-SDS-PAGE tracking the dephosphorylation of eIF2α[P] to eIF2α[0] as in **Figure 2** above, but with PP1[32 nM], G-actin [400 nM] and an escalating concentration of mutant human PPP1R15A[(533-624)V556E]. Shown is a representative of three independent

*Figure 3 continued on next page*

*Figure 3 continued*

experiments performed. The position of the mutation is provided in the schema above. The plot of initial velocity as a function of PPP1R15A$^{(533-624)V556E}$ derived from three repeats (one shown) is below the SDS-PAGE image. (**B**) As in 'A' above but using human PPP1R15A$^{(533-624)W582A}$ and G-actin [3.7 µM](Note: only the highest concentration of PPP1R15A was repeated three times). (**C**) As in 'A' above but using human PPP1R15A$^{(533-624)F592A}$ and G-actin [3.7 µM] (Note: only the highest concentration of PPP1R15A was repeated three times).

effect (*Figure 8E*, inhibition = 22% ± 2.045, unpaired t test, p<0.0001, n = 6). The weakness of salubrinal's inhibitory effect and the compound's tendency to precipitate at higher concentrations in the assay buffer frustrated our efforts to establish if inhibition was specific to the eIF2α-P directed ternary complex. Nonetheless these observations showcase the sensitivity of our assay to even weak inhibitors and strengthen the conclusion regarding Sephin1's inertness in the same assay.

## Sephin1 exerts proteostatic effects in vivo independently of PPP1R15A or the eIF2α-P-dependent integrated stress response

Sephin1's role as a proteostasis promoting agent was explored in cultured CHO-K1 cells containing reporters for both the ISR (CHOP::GFP)(*Novoa et al., 2001*) and the branch of the endoplasmic reticulum unfolded protein response (UPR) mediated by IRE1 (XBP1s::Turquoise)(*Iwawaki et al., 2004*; *Sekine et al., 2016*). Previous studies have emphasized the dominance of translational recovery in the physiological action of PPP1R15A, such that *Ppp1r15a*$^{KO}$ attenuates both the burden of protein misfolding (*Marciniak et al., 2004*) and the response to it (*Reid et al., 2016*). In keeping with these ideas and with the findings of Das and colleagues (*Das et al., 2015*), Sephin1 attenuated the activity of both UPR pathways in cells exposed to tunicamycin; an inhibitor of N-linked glycosylation, that promotes misfolding of newly-synthesized proteins (*Figure 9A*). Though observed only over a narrow concentration range of tunicamycin (*Figure 9—figure supplement 1A*) and at relatively high concentrations of the drug (*Figure 9—figure supplement 1B*), Sephin1's effects in this assay can be reconciled with a mechanism involving a net reduction of protein synthesis; as suggested by Das and colleagues. Sephin1 also inhibited induction of the ISR in response to histidinol (*Figure 9B*), an agent that interferes with tRNA charging and thereby activates the eIF2α kinase GCN2 (*Zhang et al., 2002*) without affecting protein folding.

To probe deeper into this matter, we exploited an in vivo assay that monitors eIF2α-P dephosphorylation in cells. In this kinase shut-off experiment (*Chambers et al., 2015*)(*Figure 10A*), cultured cells are first exposed to a brief, 30 min pulse of thapsigargin, which rapidly activates the eIF2α kinase PERK and builds levels of eIF2α-P, and then exposed to a PERK kinase inhibitor (GSK260414A). The resulting decay in the eIF2α-P signal reflects its dephosphorylation. To minimize the contribution of other kinases to the eIF2α-P signal, the experiment was performed in cells lacking GCN2 (*Chambers et al., 2015*), which we inactivated in the CHO-K1 cells by CRISPR-Cas9 gene editing. Normally, the dephosphorylation of eIF2α-P is a rapid process, complete in 60 min (*Figure 10B*, lanes 2–6). It was markedly delayed by inclusion of jasplakinolide (*Figure 10B* lanes 7–10), which depletes the pool of G-actin (by promoting its oligomerization), thereby depriving the PPP1R15 subunits of an essential co-factor, as observed previously (*Chambers et al., 2015*).

Together, PPP1R15A and PPP1R15B account for the bulk of eIF2α-P dephosphorylation activity of mammalian cells (*Harding et al., 2009*), but their relative contribution to the process in any given circumstance is unknown. Therefore, to adapt this assay to measure Sephin1's effect on PPP1R15A-mediated eIF2α-P dephosphorylation, it was essential to inactivate the gene encoding PPP1R15B, leaving PPP1R15A as the sole regulatory subunit of the eIF2α-P phosphatase. CRISPR-Cas9 mediated gene editing was used to create two different *GCN2*$^{KO}$; *Ppp1r15b*$^{KO}$ compound-mutant CHO-K1 cells (*Figure 10—figure supplement 1A and B*). As expected, the *GCN2*$^{KO}$; *Ppp1r15b*$^{KO}$ compound-mutant CHO-K1 cells retained their responsiveness to Sephin1 (*Figure 10—figure supplement 1C*). However, in this experimental system dependent solely on PPP1R15A, the time-dependent decline of the eIF2α-P signal (fitted to an exponential decay curve) yielded a time constant of 0.23 min$^{-1}$ for the untreated and 0.19 min$^{-1}$ for the Sephin1 treated sample, an insignificant difference (*Figure 10B* lanes 11–15, *Figure 10—figure supplement 1D and E* and *Figure 10C*).

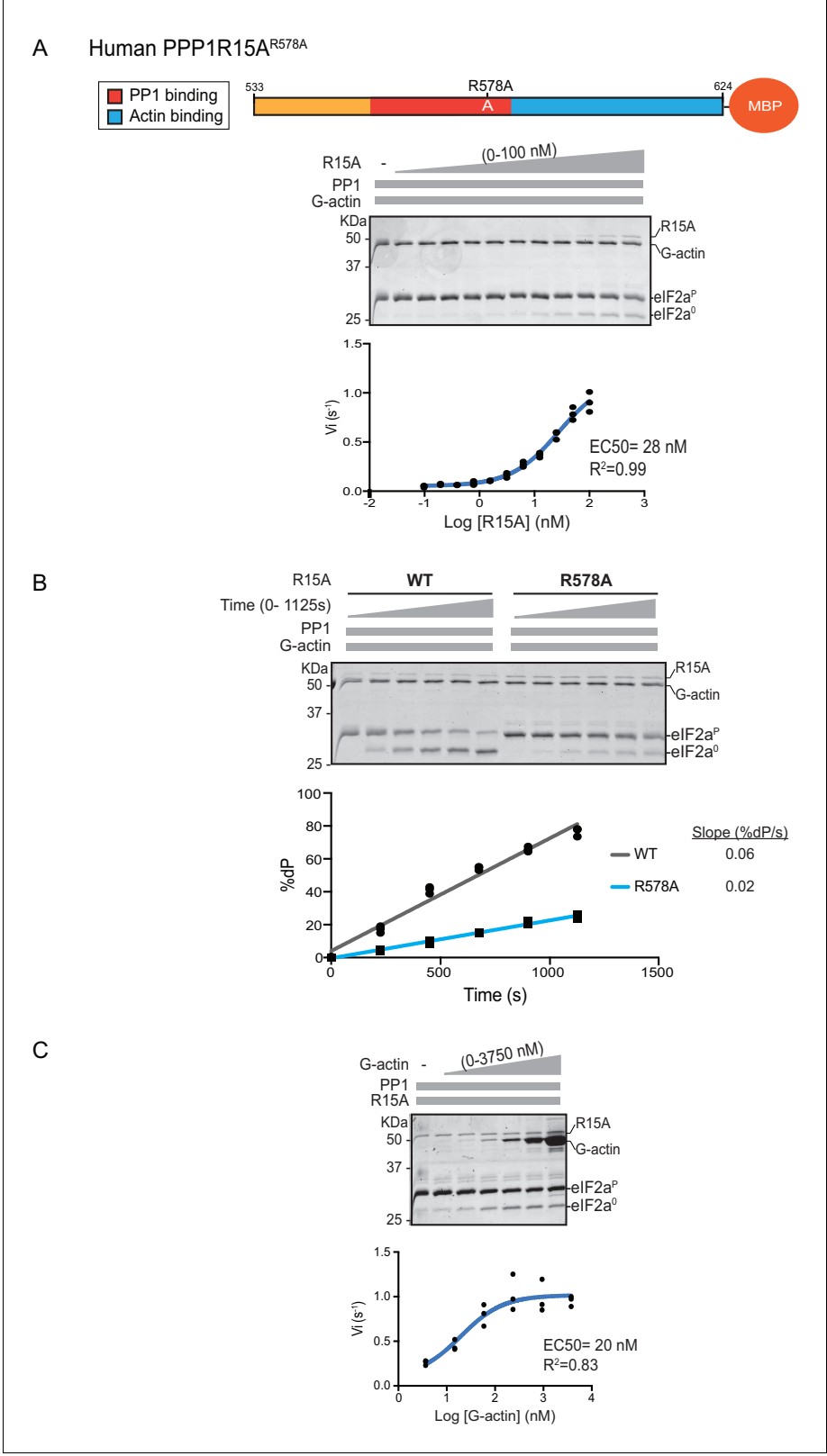

**Figure 4.** eIF2α-P dephosphorylation by ternary complexes constituted with human PPP1R15A[(533-624)R578A]. (**A**) Coomassie-stained PhosTag-SDS-PAGE tracking the dephosphorylation of eIF2α[P] in a 20 min reaction, as in *Figure 2 and 3* above, but with an escalating concentration of mutant human PPP1R15A[(533-624)R578A]. Shown is a representative of three independent experiments performed. The position of the mutation is provided in the

*Figure 4 continued*

schema above the gel. The plot of initial velocity as a function of PPP1R15A$^{(533-624)R578A}$ derived from three repeats (one shown) is below the SDS-PAGE image. The EC$_{50}$ for PPP1R15A$^{(533-624)R578A}$ was calculated using the agonist fitting curve in GraphPad Prism V7. (**B**) Time-course of eIF2α$^{P}$ dephosphorylation using a low concentration of PP1 [0.625 nM], saturating concentrations of G-actin [400 nM], and wildtype [100 nM] or mutant human PPP1R15A$^{(533-624)R578A}$ [100 nM in one assay and 200 nM in the two other assays]. Shown is a representative of three independent experiments performed. Below the gel is a plot of the fraction of substrate dephosphorylated as a function of time derived from three repeats (one shown). The slope of the reaction was derived by fitting the data to a linear model in GraphPad Prism V7. (**C**) As in 'A' above but with saturating concentration of PPP1R15A$^{(533-624)R578A}$ [100 nM] and escalating concentration of G-actin. Shown is a representative of three independent experiments performed. The plot of initial velocity as a function of G-actin derived from three repeats (one shown) is below the SDS-PAGE image. The EC$_{50}$ for G-actin was calculated using the agonist fitting curve in GraphPad Prism V7.

To mimic conditions used in the flow cytometry experiments (and those used by *Das et al., 2015*) kinase shut-off experiments were carried out on tunicamycin-treated cells (*Figure 10—figure supplement 1F*). eIF2α-P dephosphorylation proceeded rapidly in tunicamycin-treated cells. However, Sephin1 had no inhibitory effect on the rate of dephosphorylation. This experiment reveals that under conditions in which Sephin1 exerts its proteostasis-promoting activities, it does not affect rates of eIF2a-P dephosphorylation.

To follow up on this matter, both copies of the gene encoding PPP1R15A were inactivated by CRISPR-Cas9 in the reporter containing CHO-K1 cells (*Figure 11—figure supplement 1A and B*). Inactivation was confirmed by loss of the PPP1R15A signal in immunoblot of lysates from stressed mutant cells (*Figure 11A*). Sephin1 retained its ability to attenuate the response of both the XBP1s:: Turquoise and the CHOP::GFP reporter in tunicamycin treated PPP1R15A null cells (*Figure 11B*). Similar observations were made in regard to the effect of Sephin1 in histidinol-treated cells (*Figure 11C*). These observations suggest that Sephin1 also exerts its proteostatic effect(s) in cells lacking PPP1R15A.

Next, we used CRISPR-Cas9-mediated homologous recombination to introduce a site-specific mutation into the *Eif2s1* locus, to encode an ISR-blocking eIF2α$^{S51A}$mutation in the endogenous gene (*Figure 12A and B*). Surprisingly, Sephin1 retained its ability to attenuate the XBP1s::Turquoise reporter in tunicamycin-treated eIF2α$^{S51A}$mutant cells (*Figure 12C*). As CHOP activation is highly dependent on the ISR (*Harding et al., 2000*), activity of the CHOP::GFP reporter was strongly attenuated in mutant eIF2α$^{S51A}$cells. Nonetheless, it is notable that residual activation of the reporter by tunicamycin (likely a consequence of ATF6 action at the CHOP promoter [*Yoshida et al., 2000*]), was also attenuated by Sephin1 (*Figure 12C*). These observations bring into question the primacy of the ISR in Sephin1's mechanism of action.

## Discussion

The role of the eIF2α-P-dependent ISR in defending against unfolded protein stress is well supported by genetic and pharmacological experiments (*Baird and Wek, 2012*; *Ron and Harding, 2007*). By retarding its dephosphorylation, the primary consequence of eliminating PPP1R15A is to prolong the duration of the eIF2α-P signal in stress response scenarios (*Kojima et al., 2003*; *Novoa et al., 2003*) and to alter the repertoire of mRNA translation (*Reid et al., 2016*). Therefore, the finding that cells and mice lacking PPP1R15A are relatively resistant to pharmacological and genetic models associated with unfolded protein stress in the endoplasmic reticulum (ER stress) has engendered a specific interest in targeting the PPP1R15A-containing phosphatase complex for inhibition, as a means for accessing the therapeutic potential of enhanced ISR signalling.

Sephin1 and Guanabenz, compounds previously proposed to exert their proteostatic effects by disrupting the essential PP1-PPP1R15A complex and inhibiting eIF2α-P dephosphorylation (*Das et al., 2015*) are found here to have no effect in in vitro enzymatic assays dependent on the formation of a PP1-PPP1R15A complex. Sephin1 likewise proved inert in a Bio-Layer Interferometry assay that measured directly the affinity of PPP1R15A and PP1 for one another. Furthermore, we find that Sephin1 does not interfere with eIF2α-P dephosphorylation in cells (as measured by a kinase shut-off experiment) and that Sephin1 retains its ability to attenuate the impact of a challenge

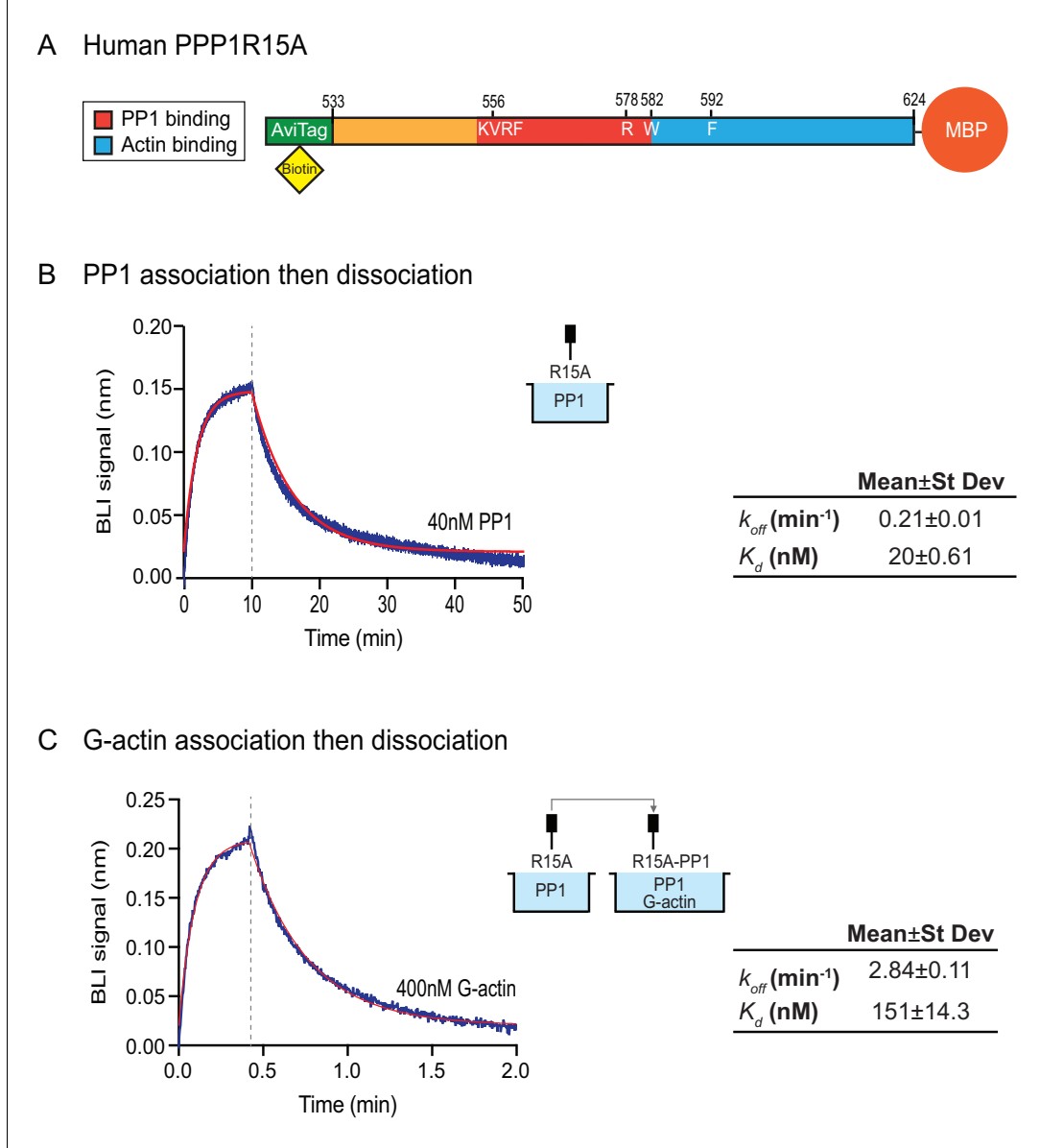

**Figure 5.** Affinity of the components of the tripartite holophosphatase for one another analysed by Bio-Layer Interferometry (BLI). (**A**) Schema of the biotinylated human PPP1R15A[533-624] immobilized onto the BLI biosensor tip. (**B**) Plot of Bio-Layer Interferometry (BLI) signal as a function of time in a representative experiment (repeated three times) in which immobilized PPP1R15A[533-624] was reacted with PP1 [40 nM] in solution (blue trace). The fitting curve using 'association then dissociation' model in GraphPad Prism V7 is shown in red. Vertical dashed line marks the beginning of the dissociation phase. Table summarizes kinetic parameters extracted from fitting curves of three repeats of the experiment shown in left panel (mean ± standard deviation). (**C**) As in 'B' above, but the immobilized PPP1R15A[533-624] was first exposed to PP1 [200 nM], before being exposed to a solution of both PP1 [200 nM] and G-actin [400 nM]. Shown is a representative of an experiment repeated three times.

to proteostasis even in cells lacking PPP1R15A, or in ISR-defective *Eifs1*[S51A] cells. These observations suggest that the previously-reported attenuation of the recovery of PP1 in complex with PPP1R15A, when both were purified from lysates of cells treated with Sephin1 was unlikely to be a direct consequence of Sephin1 interference with complex formation or of destabilization by Sephin1 of a pre-existing complex and also call into question the importance of any indirect disruption of the PP1-PPP1R15A complex that may occur in vivo and remain undetected by our assays.

Our findings, questioning whether Sephin1 attains its proteostatic activity by inhibiting the PPP1R15A-containing eIF2α-P directed holophosphatase, and similar concerns raised by the Peti

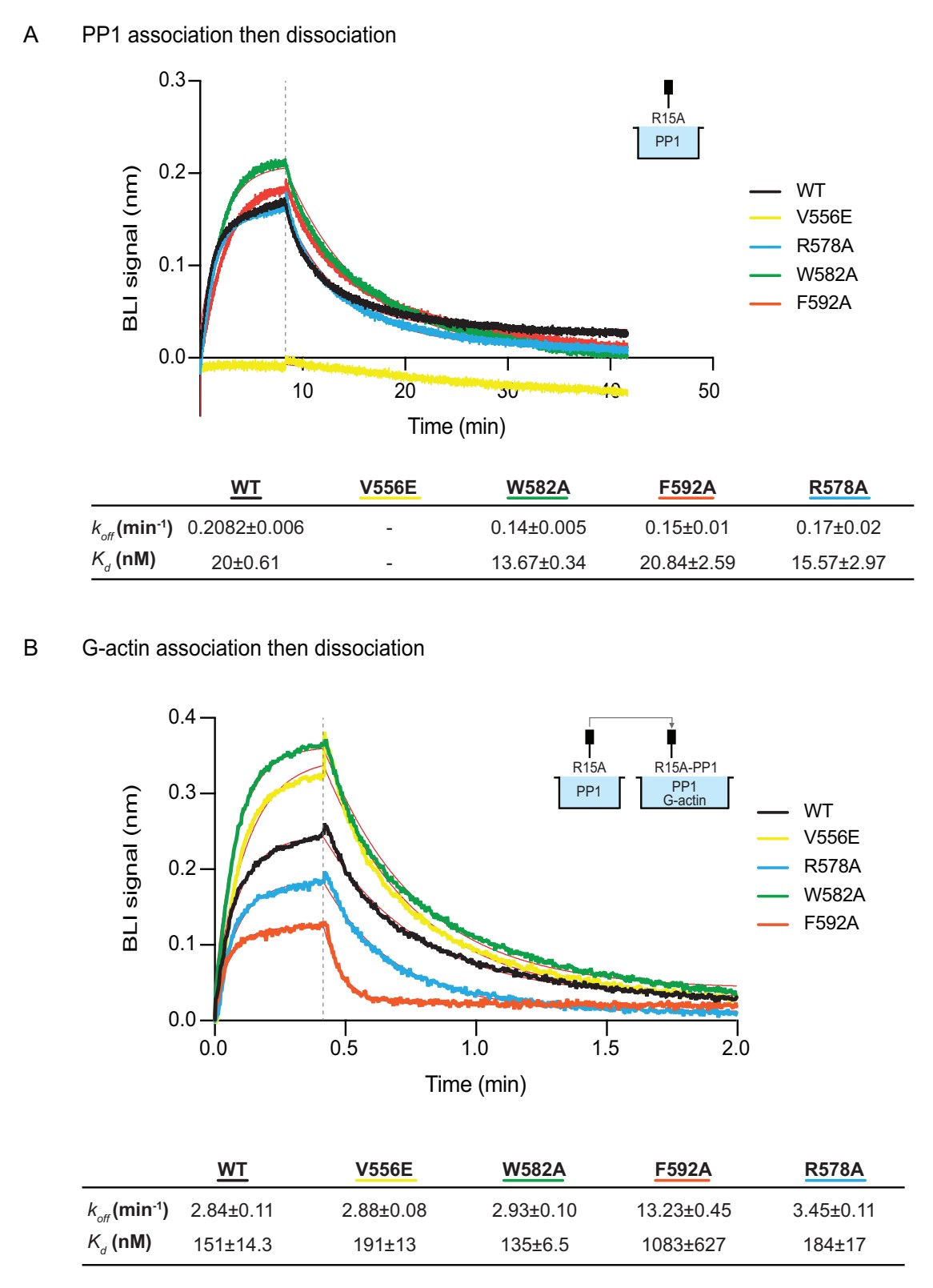

**Figure 6.** The effect of human PPP1R15A mutations on the affinity of the components of the tripartite holophosphatase for one another analysed by Bio-Layer Interferometry (BLI). (**A**) Plot of Bio-Layer Interferometry (BLI) signal as a function of time in a representative experiment (repeated three times) in which immobilized wildtype and indicated mutant PPP1R15A[533-624] proteins were reacted with PP1 [40 nM] in solution (thick traces). The fitting curve using 'association then dissociation' model in GraphPad Prism V7 is shown in thin red line. Vertical dashed line marks the beginning of the dissociation

*Figure 6 continued on next page*

*Figure 6 continued*

phase. Table summarizes kinetic parameters extracted from fitting curves of three repeats of the experiment shown in left panel (mean ± standard deviation). (**B**) As in 'A' above, but the immobilized wildtype and mutant PPP1R15A[533-624] probes were first reacted with PP1 [200 nM], before being exposed to a solution of both PP1 [200 nM] and G-actin [400 nM]. Shown is a representative of an experiment repeated three times.

and Shenolikar labs (*Choy et al., 2015*), do nothing to diminish the attractiveness of PPP1R15A inhibition as a potential means for defending proteostasis. Similarly, there is nothing in our study to question the beneficial effects reported for Sephin1 in mouse models of neurodegeneration (*Das et al., 2015*) nor we do not challenge the enhanced susceptibility of *Ppp1r15b*[KO] cells to the [(*o*-chlorobenzylidene)amino]guanidine, Guanabenz (*Tsaytler et al., 2011*). However, our findings that Sephin1 exerts its effects in CHO-K1 cells lacking PPP1R15A or in ISR-defective *Eifs1*[S51A] cells raise doubts as to whether these phenomena were attained via inhibition of PPP1R15A or indeed modulation of the ISR.

Crystal structures of the PPP1R15(A or B)-PP1 and the related PNUTS/PPP1R10-PP1 and spinophilin/PPP1R9B-PP1 complexes reveal that both the residues corresponding to human PPP1R15A[V556] (of the RVxF motif) and the conserved arginine (human PPP1R15A[R578]) insert deeply into the surface of the PP1 subunit (*Chen et al., 2015*; *Choy et al., 2014*, *2015*). The contrast between the dramatic effect of the PPP1R15A[V556E] mutation and the more modest effect of the human PPP1R15A[R578A] mutation on the PPP1R15A-PP1 complex may reflect a role for the former early in the pathway to complex assembly (a process that can be thought of as PPP1R15A folding on the surface of PP1). It is therefore possible that inhibitors of the eIF2α-P holophosphatase might disrupt complex assembly, without affecting the stability or activity of a preformed complex. However, Sephin1 is unlikely to belong to such a category, as it failed to exert an inhibitory effect on enzymatic activity or on the BLI signal even when added to pure PPP1R15A, before addition of PP1 and G-actin. But other compounds that remain to be found might selectively disrupt the assembly of the PPP1R15A-PP1 complex by binding to and stabilizing an intermediate step in its formation.

Similarly instructive is the human PPP1R15A[W582A] mutation, which eliminates all detectable selectivity of PPP1R15-holophosphatases for eIF2α-P (*Chen et al., 2015*)(and *Figure 3B* here) without affecting the kinetics of the bimolecular association of PPP1R15A with PP1 or G-actin. These features suggest that the side chain of W582 - a residue conserved throughout the PPP1R15 family - may have a special role in aligning PP1 and G-actin to form a composite surface with affinity for the substrate, without contributing measurably to the stability of the tripartite holophosphatase. That the side chain of single tryptophan residue can bias the complex towards activity (without affecting its stability), suggests the possibility that small molecules might access this allosteric mechanism and bias the tripartite holophosphatase towards or away from enzymatic activity without the need to disrupt an extensive protein-protein interface.

Specific cellular dephosphorylation events are notoriously difficult to target with small molecules (*Sakoff and McCluskey, 2004*). Here we presented evidence that the perception of Sephin1 as a milestone in overcoming that challenge may need re-thinking. However, features of the PPP1R15A-PP1-G-actin holophosphatase noted above suggest ways in which eIF2α-P dephosphorylation might indeed be selectively targeted by small molecules. In vitro assays for selective eIF2α-P dephosphorylation, such as the one described here, might prove useful in discovery of small molecules with such a mechanism of action.

## Materials and methods

### Plasmid construction

Diverse cloning techniques were used to create the bacterial and mammalian expression vectors listed in *Table 1*. This table contains information about lab number, name, description and reference for each plasmid used.

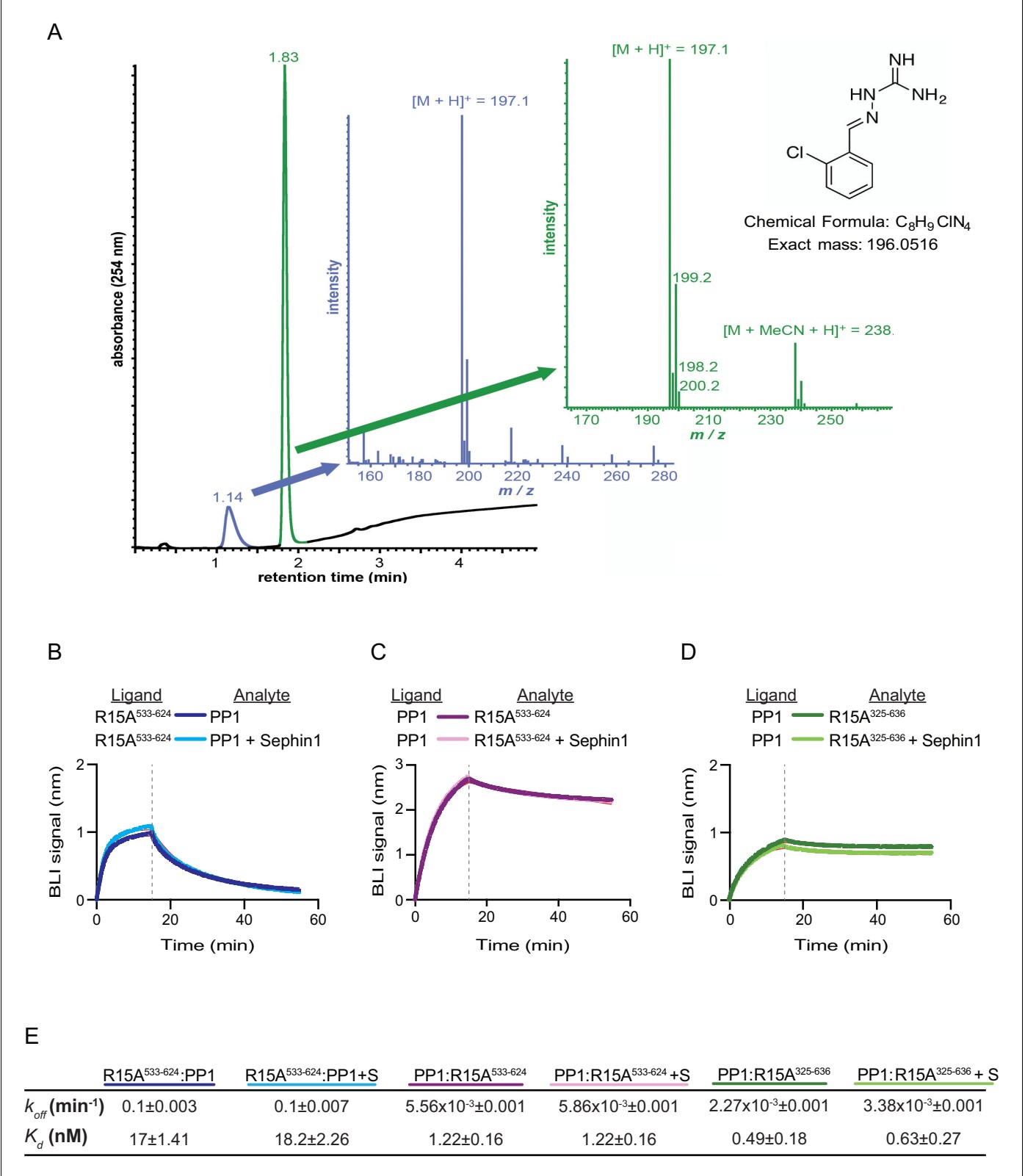

**Figure 7.** Sephin1's effect on PP1-human PPP1R15A association analysed by Bio-Layer Interferometry (BLI). (**A**) From left to right: Absorbance trace (at 254 nm) of Sephin1 resolved by reverse-phase HPLC and mass spectra of the minor early (blue) peak eluting at 1.14 min and the major later eluting (green peak) at 1.83 min. The predicted structure, chemical formula and exact mass of Sephin1, are provided for reference. (**B**) Plot of BLI signal as a function of time of a representative experiment (repeated three times) performed with immobilized PPP1R15A[533-624] (Ligand) and PP1 [40 nM] (Analyte). *Figure 7 continued on next page*

*Figure 7 continued*

Where indicated, the analyte was mixed with Sephin1 [50 µM] which was then present during the binding phase of the experiment. (C) As in 'B' above but with biotinylated PP1 as the ligand and PPP1R15A[533-624] as the analyte, in the absence or presence of Sephin1. (D) As in 'B' above but with biotinylated PP1 as the ligand and PPP1R15A[325-636] as the analyte, in the absence or presence of Sephin1. (E) Table summarizing data extracted from fitting curves of three repeats of the experiments shown above (mean ± standard deviation).

## Protein expression and purification

Actin was purified from rabbit muscle according to (*Pardee and Spudich, 1982*) as modified by (*Chen et al., 2015*).

Expression plasmids for PPP1R15A (GADD34) variants contained ampicillin resistance marker, N-terminal GST tag and C-terminal maltose binding protein (MBP) tag (UK1677, UK1920)(*Table 1*). They were transformed into BL21 T7 Express lysY/Iq *E. coli* (C3013, New England Biolabs) and colonies that grew in LB-ampicillin plates (100 µg/ml ampicillin) were used to create a saturated overnight culture. This saturated culture was used to inoculate 2–4 Litres of LB media supplemented with 100 µg/ml ampicillin. The cultures were incubated at 37°C until OD600 = 0.6–0.8. At this point, they were induced with 1 mM Isopropyl $\beta$-D-1-thiogalactopyranoside (IPTG) and cultured for 20 more hours at 18°C. It was followed by a centrifugation step to pellet bacteria and resuspension of the ice-cold pellets in 3–4 pellet volumes of lysis buffer (50 mM Tris pH 7.4, 500 mM NaCl, 1 mM $MnCl_2$, 1 mM $MgCl_2$, 1 mM tris(2-carboxyethyl)phosphine (TCEP), 100 µM phenylmethylsulfonyl fluoride (PMSF), 20 mTIU/ ml aprotinin, 2 µM leupeptin, and 2 µg/ml pepstatin in 10% glycerol). An Emulsi-Flex-C3 homogenizer (Avestin, Inc, Ottawa, Ontario) was used to lyse the bacteria, which were then clarified in a JA-25.50 rotor (Beckman Coulter) at 33,000×g for 30 min at 4°C. These suspensions were bound to pre-equilibrated glutathione sepharose 4B beads (17-0756-05, GE Healthcare) for 1–2 hr at 4°C. Beads were transferred to a 10 mL column after being batch-washed with 20 bed volumes of lysis buffer. Proteins were eluted in glutathione elution buffer (50 mM Tris pH 7.4, 100 mM NaCl, 40 mM glutathione, 0.5 mM $MnCl_2$, 0.5 mM TCEP, 10% glycerol), and cleaved with Tobacco Etch Virus protease (TEV) (12.5 µg TEV protease/mg protein) overnight at 4°C to remove the N-terminal GST tag. Cleaved proteins were bound to amylose beads (E8021S, New England Biolabs) for 1–2 hr at 4°C. Twenty/thirty bed volumes of lysis buffer were used to batch-wash the amylose beads, which were transferred to a 10 mL column and eluted with HEPES buffer (20 mM HEPES, 100 mM NaCl, 0.2 mM $CaCl_2$, 0.2 mM ATP, 0.2 mM TCEP, 0.5 mM $MnCl_2$, 100 µM PMSF, 20 mTIU/ ml aprotonin, 2 µM leupeptin, and 2 µg/ml pepstatin) and 10 mM maltose.

PP1 (UK622) (*Table 1*) was purified as above, with the following modifications: LB media cultures were supplemented with $MnCl_2$, after TEV cleavage proteins were buffer exchanged using a 2 mL desalting column in HEPES buffer and re-bound to glutathione sepharose 4B beads to remove free GST tag.

Phosphorylated eIF2$\alpha$ was encoded by an expression plasmid containing N-terminal His-Tag and kanamycin resistance marker (UK105) (*Table 1*). BL21 T7 Express lysY/Iq E. coli were co-transformed with this plasmid and a GST-Tagged PERK plasmid carrying ampicillin resistance marker (UK168) (*Table 1*). Colonies that grew in ampicillin (100 µg/ml) and kanamycin (50 µg/ml) LB-plates were used to create a saturated over-night cultured with which 2L of ampicillin and kanamycin LB were inoculated. Growth, induction and purification was as described for PPP1R15A, with the following changes: beads used were Ni-NTA (30230, Qiagen) to bind His-tag, lysis buffer contained 20 mM imidazole and elution buffer contained 500 mM imidazole instead of glutathione. This protein did not require TEV cleavage but an additional size exclusion chromatography step was included. A Superdex S200 (GE Healthcare) was used to gel filter the protein in 25 mM Tris, 100 mM NaCl, 0.1 mM EDTA, 1 mM DTT and 10% glycerol buffer.

All proteins were snap frozen and kept at −80°C in small aliquots. Final concentration of proteins was calculated from UV absorbance at 280 nm measurements in Nanodrop (Thermo Scientific, UK) and based on their extinction coefficient values predicted by MacVector.

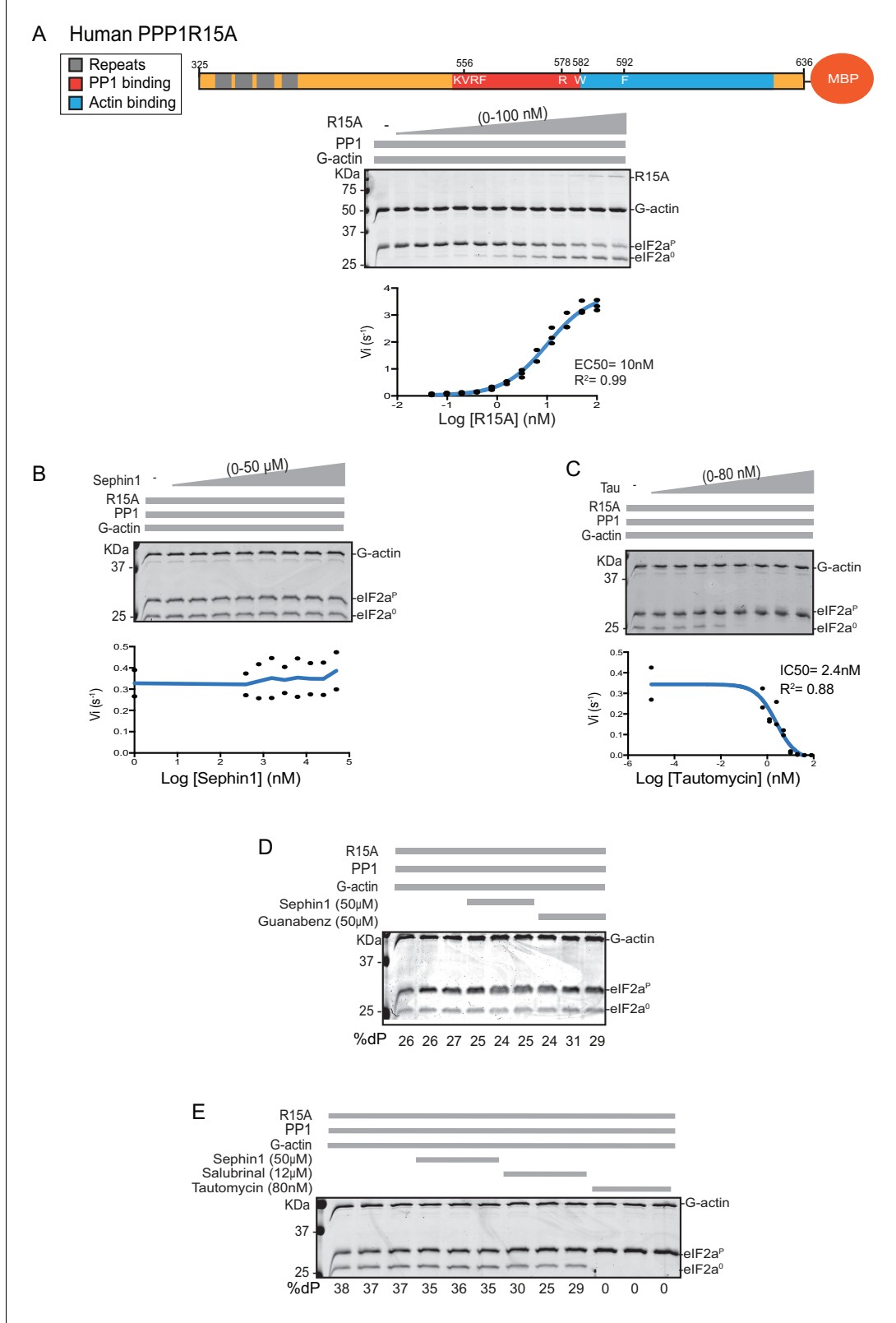

**Figure 8.** Sephin1's effect on the eIF2α-P dephosphorylation activity of the human PP1-PPP1R15A-G-actin holophosphatase in vitro. (**A**) Coomassie-stained PhosTag-SDS-PAGE tracking the dephosphorylation of eIF2α$^P$ in 20 min reactions constituted with PP1 [0.625 nM], G-actin [1.5 μM] and an escalating concentration of human PPP1R15A$^{325-636}$. Shown is a representative of three independent experiments performed. A schema of the human PPP1R15A$^{325-635}$ construct is shown above the gel. A semi-log$_{10}$ plot of the initial velocity of eIF2α$^P$ dephosphorylation as a function of PPP1R15A$^{325-636}$

*Figure 8 continued on next page*

*Figure 8 continued*

concentration derived from three repeats of the experiment is shown below. The EC$_{50}$ for PPP1R15A$^{325-636}$ was calculated using agonist fitting function on GraphPad Prism V7. (B) As in 'A' above, but in the presence of a fixed concentration of PPP1R15A$^{325-636}$ below the EC$_{50}$ [2 nM] and escalating concentrations of Sephin1. Shown is a representative of the two independent experiments performed. Plot contains data from the two repeats. (C) As in 'B' above, but in the presence of an escalating concentrations of the PP1 active site inhibitor tautomycin (Tau). Shown is a representative of the two independent experiments performed. Plot contains data from the two repeats. (D) As above, triplicate reactions of eIF2α-P dephosphorylation conducted in the absence or presence of Sephin1 or the related compound, Guanabenz. (E) As in 'D' using Sephin1, salubrinal or tautomycin. Shown is a representative experiment, (of two repeats).

The following figure supplement is available for figure 8:

**Figure supplement 1.** Sephin1's effect on the eIF2α-P dephosphorylation activity of the mouse PP1-PPP1R15A-G-actin holophosphatase in vitro.

## In vitro biotinylation reactions

Biotin (B1595, Thermo Scientific) was added to the AviTagged specified proteins (encoded by UK1897, 1920, 1921, 1992, 1993, 1994, 1995)(*Table 1*) using BirA. BirA was amplified by PCR reaction from *E. coli* genomic DNA and inserted into an expression vector containing N-terminal GST (TEV cleavable) and ampicilin resistance marker (UK1881) (*Table 1*). This protein was purified following standard GST-tagged protocols, eluted in glutathione elution buffer, aliquoted and stored in this buffer. All proteins were biotinylated and its biotinylation was checked as described (*Fairhead and Howarth, 2015*)

Proteins in glutathione elution buffer were biotinylated and buffer exchanged into a HEPES buffer to remove excess of biotin that would interfere with the Bio-Layer Interferometry measurements.

## In vitro dephosphorylation of eIF2αP

**Drugs used:** Sephin1 (EN300-195090, Enamine), tautomycin (5805551, Calbiochem), Guanabenz (D6270, Sigma-Aldrich), salubrinal (Sal003, S4451, Sigma-Aldrich)

Dephosphorylation reactions were conducted as described (*Chen et al., 2015*). In summary, reactions were conducted by combining the different proteins for 20 min at 30°C whilst shaking at 500 rpm and were stopped by addition of Laemmli buffer. A fraction of the reactions were loaded into PhosTag SDS-PAGE, stained using Coomasie and scanned. ImageJ (NIH) was used to quantify signal intensity.

Enzyme velocity, *V* was measured at substrate concentrations well below the enzyme's $K_m$ and in samples with less than 25% substrate depletion. Under these conditions, the instantaneous velocity (i.e. rate of substrate conversion to product per molecule of enzyme) is proportional to instantaneous substrate concentration and the equivalent velocity is obtained with the equation below, derived from the integrated rate equation for first order kinetics:

$$Vi = \frac{ln\frac{[S]_0}{[S]_f} * [S]_0}{\Delta t * [ENZ]}$$

Where $Vi$ is the initial velocity (the instantaneous velocity at t = 0, with the dimensions of 1/t), $[s]_0$ and $[s]_f$ are, respectively, the substrate concentrations at the beginning and end of the reaction, $\Delta t$ is the time interval of the reaction and $[ENZ]$ is the concentration of enzyme. The 'agonist fitting' or 'inhibitor fitting' functions of GraphPad Prism V7 (RRID: SCR_002798) were used to analyze the effects of varying concentrations of reaction components on velocity.

## Bio-Layer Interferometry (BLI) measurements

Proteins were diluted in HEPES buffer at the specified concentrations. Two-hundred microliters of each diluted protein preparation was placed in a 96 well plate (655209, Greiner). Streptavidin sensors (18–5019, ForteBio) were hydrated in this buffer for 2–5 min before the binding assay was performed. The plate was placed in the ForteBio Octet RED96 System for data acquisition which was performed at 25°C at a constant orbital flow of 600 rpm. The binding assays consisted of the measurement of change in layer thickness (in nanometres) during a series of sequential steps. The sensor was equilibrated in the buffer (240 s) and the ligand (biotyninated protein) was loaded on the sensor.

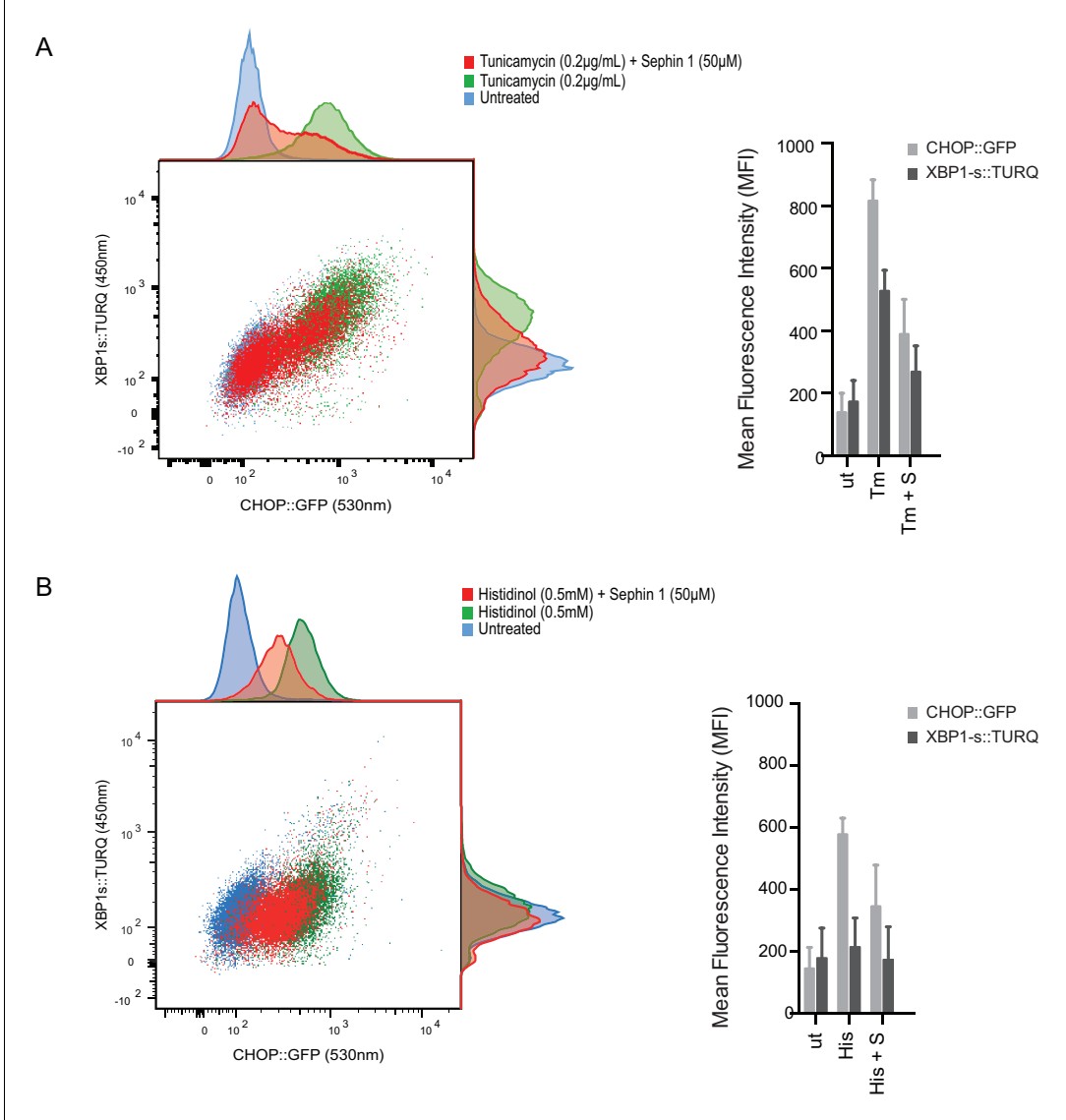

**Figure 9.** Sephin1 broadly attenuates the ER stress response in cultured CHO cells. (**A**) Two-dimensional plot of the fluorescence signals derived from CHO cells stably transduced with both a CHOP::GFP reporter (on the horizontal axis, Ex: 488 nm/ Em 530 ± 30 nm; reflecting mostly ISR activity) and a XBP1::Turquoise reporter (on the vertical axis, Ex: 405 nm/ Em 450 ± 50 nm; reflecting IRE1α activity) analysed by flow cytometry. Color-coded signals from untreated cells (blue) or cells exposed to a low concentration of tunicamycin (0.2 µg/mL; 20 hr) alone (green) or together with Sephin1 (50 µM, red) are superimposed. Histograms of the distribution of the two reporter signals in the three cell populations are plotted on the corresponding axis and the mean ± CV (coefficient of variation) of the fluorescence intensity of the two reporters is depicted in the bar diagram to the right. (**B**) As in 'A' above, but the cells were exposed to histidinol, an ISR inducer that does not promote unfolded protein stress in the ER and does not activate the XBP1:: Turquoise reporter. Shown is one of three independent experiments.

The following figure supplement is available for figure 9:

**Figure supplement 1 .** Concentration-dependence of the response of cultured cells to tunicamycin and Sephin1.

Ligand attachment to the sensor was checked by immersion of the sensor in buffer after loading (400–2000 s). Finally, association and dissociation of the proteins studied was analysed by soaking the sensor in analyte solutions and buffer, respectively. The duration of the ligand loading on the sensor was set to a specific time (600 s) or a specific value (2 nm displacement) depending on the experiment performed. The duration of the association and dissociation of the analyte to the ligand,

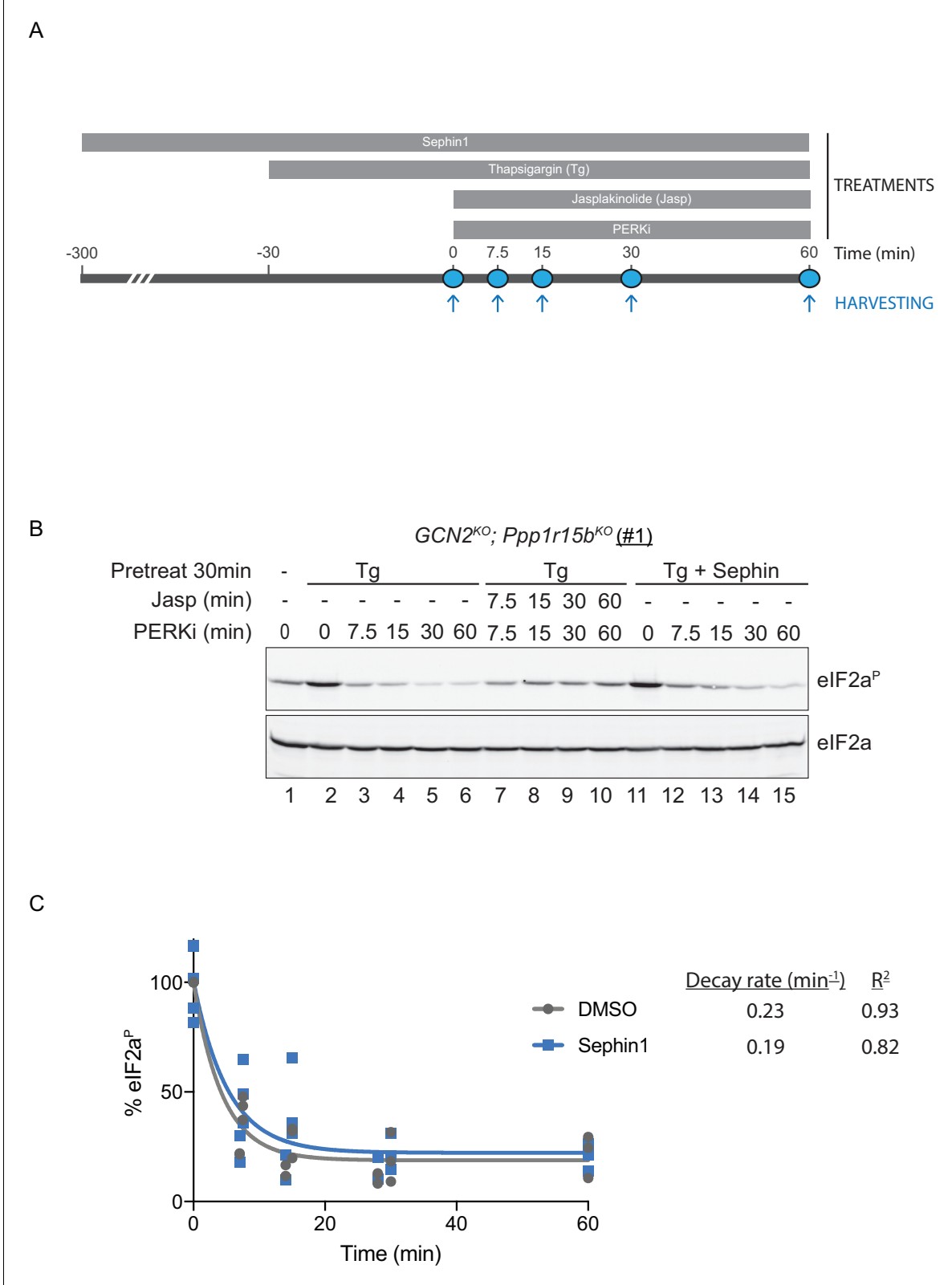

**Figure 10.** eIF2α$^P$ dephosphorylation in untreated and Sephin1-treated cells. (**A**) Schema of the kinase shut-off experiment used to evaluate the decay of the eIF2α$^P$ signal in cells. Thapsigargin (300 nM) was added at t = −30 min to the media to activate PERK kinase and induce eIF2α phosphorylation. Sephin1 (50 μM) was introduced either at t = −30 min (alongside thapsigargin, in the experiment shown in panel B below and in *Figure 10—figure supplement 1* panel C) or at t = −300 min (*Figure 10—figure supplement 1* panel D). A PERK kinase inhibitor, PERKi/GSK260414A (2 μM), was added
*Figure 10 continued on next page*

Figure 10 continued

at t = 0 to visualize eIF2α$^P$ dephosphorylation at specified times. (**B**) Immunoblot of the time-dependent changes in the eIF2α$^P$ signal of compound-mutant *Ppp1r15b*$^{KO}$; *Gcn2*$^{KO}$ CHO-K1 cells (clone #1) treated as in 'A'. Where indicated, the cells were additionally exposed to Sephin1 (50 μM) or the actin-polymerizing agent Jasplakinolide (1 μM), which inhibits eIF2α$^P$-dephosphorylation by sequestering G-actin. The immunoblot of eIF2α (lower panel) serves as a loading control. Shown is a representative experiment repeated five times. (**C**) Plot of the eIF2α$^P$ signal (normalised to the value at t = 0 of the vehicle only (DMSO) sample) as a function of time derived from five independent experiments. The data have been fitted to an exponential decay curve (grey solid line for the vehicle and blue solid line for the Sephin1-treated sample). The exponential decay rate and the R$^2$ of the fit are indicated.

The following figure supplement is available for figure 10:

**Figure supplement 1.** Analysis of Sephin1 in a different *Ppp1r15b* mutant cell line.

was adjusted in order to capture bindings that had not reached equilibrium phase. Data analysis were performed using GraphPad V7 (RRID: SCR_002798) and curves were fitted to a receptor binding kinetics association then dissociation built-in model.

## Gene editing

### Ppp1r15a mutant cells

Dual reporter CHOP::GFP, XBP1::Turquoise CHO-K1 cell line (clone S21 a derivative of RRID: CVCL_0214) (**Sekine et al., 2015**) were chosen to create *Ppp1r15a* knock out clones by CRISPR/Cas9 system (**Ran et al., 2013**). The identity of the S21 cells and their mutant derivatives has been confirmed by the persistence of the CHOP::GFP marker introduced into CHO-K1 cells (originally obtained from ATCC, catalogue number CCL-61) by the presence of proline auxotrophy and by genomic sequencing, which confirms them to be of *Cricetulus griseus* origin. Mycoplasma contamination is monitored frequently in our cell culture facility by cytoplasmic DAPI staining and by PCR

CRISPy database (URL: http://staff.biosustain.dtu.dk/laeb/crispy/) was used to select single guide RNA sequences to target the PPP1R15A-encoding gene in exon 1 (upstream the PP1 binding motif). The two sequences selected were CRISPy Target ID 1668683 and 1671391 and duplex DNAs of the sequences were inserted into the pCas9-2A-GFP (UK1359) (**Table 1**) plasmid to create CHO_PPP1R15A_guide1_pSpCas9(BB)−2A-GFP (UK1599) and CHO_PPP1R15A_guide2_pSpCas9(BB)−2A-GFP (UK1600),(**Table 1**), respectively.

CHO-K1 cells were transfected with either plasmid (UK1599 or UK1600) (**Table 1**) using Lipofectamine LTX (Invitrogen). Twenty-four hours later, cells were washed with PBS and resuspended in PBS containing 4 mM EDTA. A MoFlo Cell Sorter (Beckman Coulter) was used to individually sort GFP-positive cells (confirming plasmid transfection). Genomic analysis of the clones was performed using a PCR-based assay. Primers were designed to amplify the *Ppp1r15a* region targeted by the RNA guides. The reverse primer was labelled with 6-carboxyfluorescein (6-FAM) on the 5' end, to create fluorescently-labelled PCR products. The diluted PCR products were loaded on a 3130xl Genetic Analyzer (Applied Biosystems) and analysed using the Gene Mapper software (Applied Biosystems) to determine their length. Clones in which frame-shifting mutations were predicted by size of the PCR product, were sequenced. PPP1R15A$^{KO}$(clone #1) (from Guide 1) was identified as compound heterozygous for two gene-disrupting alleles [1479_1492delGCTCAGGGTTGTCT/1491_1492ins(440n)] and PPP1R15A$^{KO}$ (clone #2) (from Guide 2) as homozygote [1588_1589insA]. All three alleles have a frame shift mutation 5' of the PP1 binding motif with no intervening AUG codon for in frame down-stream translation initiation of a fragment containing the PP1 binding motif.

### Eif2ak4 (GCN2); Ppp1r15b compound-mutant cells

Same dual reporter CHOP::GFP, XBP1s::Turquoise CHO cell line (clone S21) was chosen to create compound-mutant GCN2$^{KO}$; Ppp1r15b$^{KO}$ knock out clones by CRISPR/Cas9 system.

A CRISPR guide was designed to target the region of exon 9 of *Eif2ak4 (GCN2)* that is upstream of the kinase domain. As previously described, a duplex DNA of CRISPy Target ID 1051489 was introduced in pCas9-2A-puro plasmid to create CHO_EIF2K4_guideA_pSpCas9(BB)−2A-Puro (UK1497) (**Table 1**). Cell sorting was based on loss of the ISR (lost of CHOP::GFP signal) upon Histidinol treatment. The *Eif2ak4* genomic region from sorted cells was sequenced.

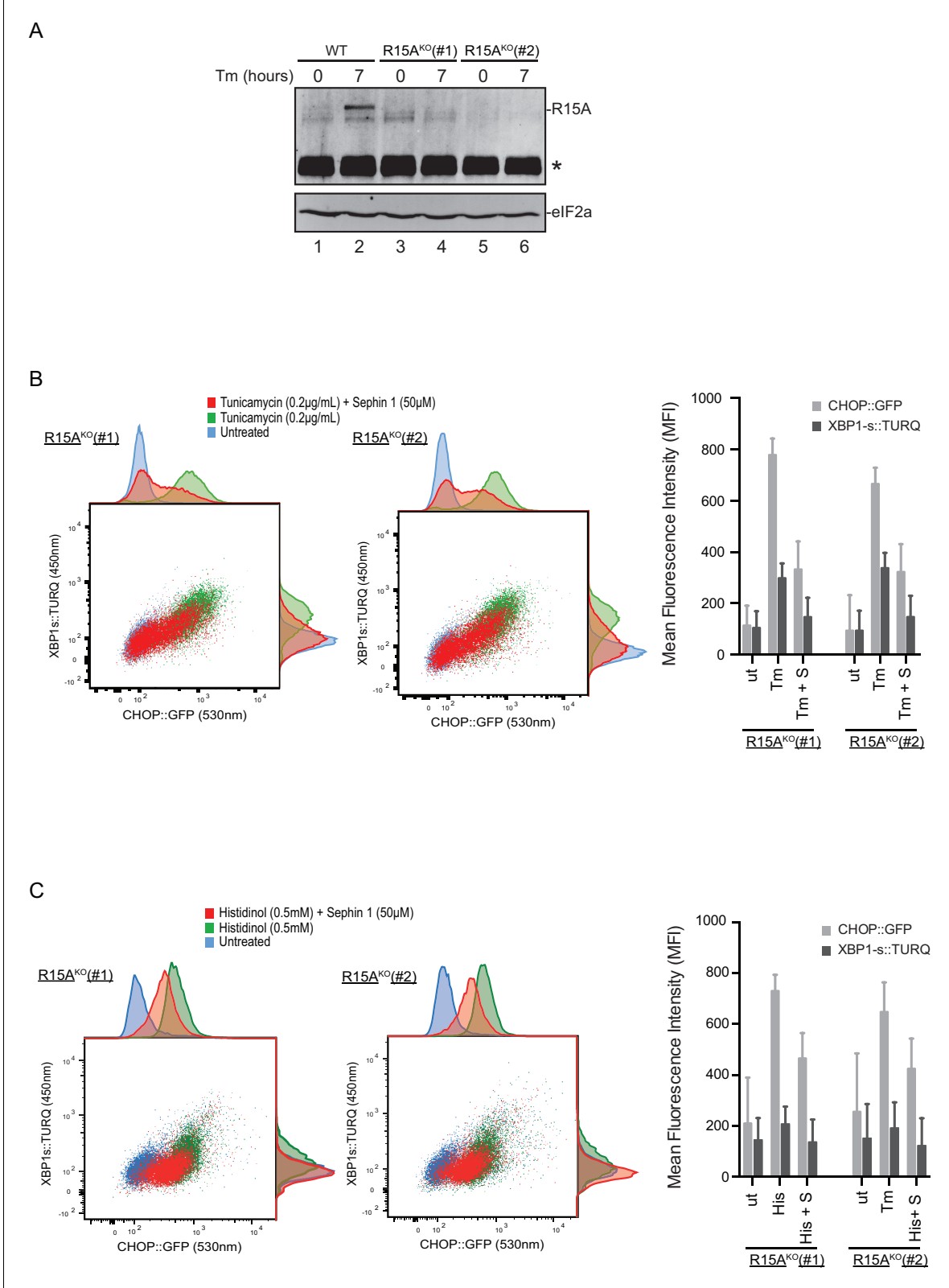

**Figure 11.** Cells lacking PPP1R15A remain responsive to Sephin1. (**A**) Immunoblot of endogenous PPP1R15A recovered by immunoprecipitation (using an anti-PPP1R15A antibody conjugated to Protein A Sepharose) from untreated and tunicamycin exposed parental cells and cells from two different *Ppp1r15a*[KO] CHO-K1 clones. The position of PPP1R15A is indicated and the immunoglobulin heavy-chain is marked with an asterisks. The immunoblot of eIF2α (lower panel) serves as a loading control for the content of cellular protein in the lysates. (**B**) Two-dimensional plot and histograms of the

*Figure 11 continued on next page*

*Figure 11 continued*

fluorescent signal of the CHOP::GFP and XBP1s::Turquoise reporters in the two *Ppp1r15a*^KO CHO-K1 clones. Where indicated, the cells were exposed to a low concentration of tunicamycin (0.2 µg/mL; 20 hr) alone or together with Sephin1 (50 µM). The mean ± CV (coefficient of variation) of the fluorescence intensity of the two reporters in each of the two clones is displayed in the bar diagram. Shown is one of three independent experiments. (C) As in 'C' above, but cells were exposed to histidinol. Shown is one of three independent experiments.

The following figure supplement is available for figure 11:

**Figure supplement 1 .** Mutant *Ppp1r15a* alleles.

The GCN2^KO clone selected was identified as homozygous for a 36019_36020insG InDel that encodes a truncated protein lacking the kinase domain.

CRISPR guides were designed to target *Ppp1r15b* at either the 5' end (Guide A, CRISPy Target ID 1315067) or 3' end (Guide B, CRISPy Target ID 1315106) of exon1; both upstream of the catalytic KVxF motif spanning the junction of exons 1 and 2. Guide DNA duplexes for each target were introduced into pSpCas9(BB)−2A-mCherry_V2 (UK1610) (*Table 1*) to create CHO_PPP1R15B_guideA_pSpCas9(BB)−2A-mCherry_V2 (UK2081) and CHO_PPP1R15B_guideB_pSpCas9(BB)−2A-mCherry_V2 (UK2082) (*Table 1*). Positive CHO-K1 cell transfectants were sorted for mCherry expression by FACS. The *Ppp1r15b* genomic region from sorted cells was sequenced.

The selected clone targeted by guide A (clone #1) was heterozygous for a 45_52del and 48_49insC. The selected clone targeted by guide B (clone #2) was heterozygous for a 1724del and 1722_1725del (residue numbering based on NCBI Reference Sequence: NW_003614184.1). Both clones encode only truncated PPP1R15B proteins, each lacking the KVxF containing catalytic domain.

(note: the existing anti-PPP1R15B sera do not recognize the hamster protein, hence confirmation of gene disruption was confined to genotypic analysis which revealed frame-shifting that precludes expression of the active C-terminal fragment).

## Eif2s1 (eIF2α) gene editing

High fidelity homology directed repair (HDR) CRISPR/Cas9 system was used to create Ser51Ala mutation in eIF2α in dual reporter CHOP::GFP, XBP1s::Turquoise CHO-K1 cell line (clone S21).

Duplex DNAs of guide sequence [CRISPy Target ID: 1051485] was inserted into pCas9-2A-puro vector to create CHO_Eif2s1_guideC_pSpCas9(BB)−2A-Puro (UK1507) (*Table 1*). CHO cells were transfected with this plasmid and a 190 bp single-stranded DNA oligonucleotide (ssODN) that carried the desired mutation (Ser51Ala) and a PAM mutation (to abolish the Cas9 cleavage site in recombinant alleles). Cells that were CHOP::GFP negative upon histidinol treatment were single-cell sorted. The genotypic analysis of the selected clone showed that it is heterozygous, one allele contains the desired mutation [5307_T>G (Serine), 5321_C>T (PAM)] and the other allele has an insertion that produces a truncated version of the protein (5326_5327insT), thus the only functional copy of eIF2α in this cell has the S51A mutation.

## Flow cytometry analysis

CHO cells were plated in six well plate at $3 \cdot 10^5$ cells/well density. Next day, they were treated for 20 hr with specified compounds. They were washed twice with PBS and suspended in PBS 4 mM EDTA to be evaluated by flow cytometry. Flow cytometry data were analyzed using FlowJo (FlowJo,LLC, RRID: SCR_008520) and GraphPad-Prism V7 (RRID: SCR_002798) was used to create bar graphs.

Replicates of flow cytometry experiments were analysed using Stata 14 *StataCorp. (2015. Stata Statistical Software: Release 14. College Station, TX: StataCorp LP*, RRID: SCR_012763). The interaction between treatments and genotypes in the different repeats was modeled using linear regression. The model was used to test whether the effect of different treatments differed between the genotypes, allowing for different mean values of CHOP::GFP and XBP1s::Turquoise on each repeat. The analysis showed non-significant differences between genotypes. However, for each cell type, there were significant differences (p<0.02) between untreated cells versus stressed cells and also between the latter and cells co-treated with Sephin1.

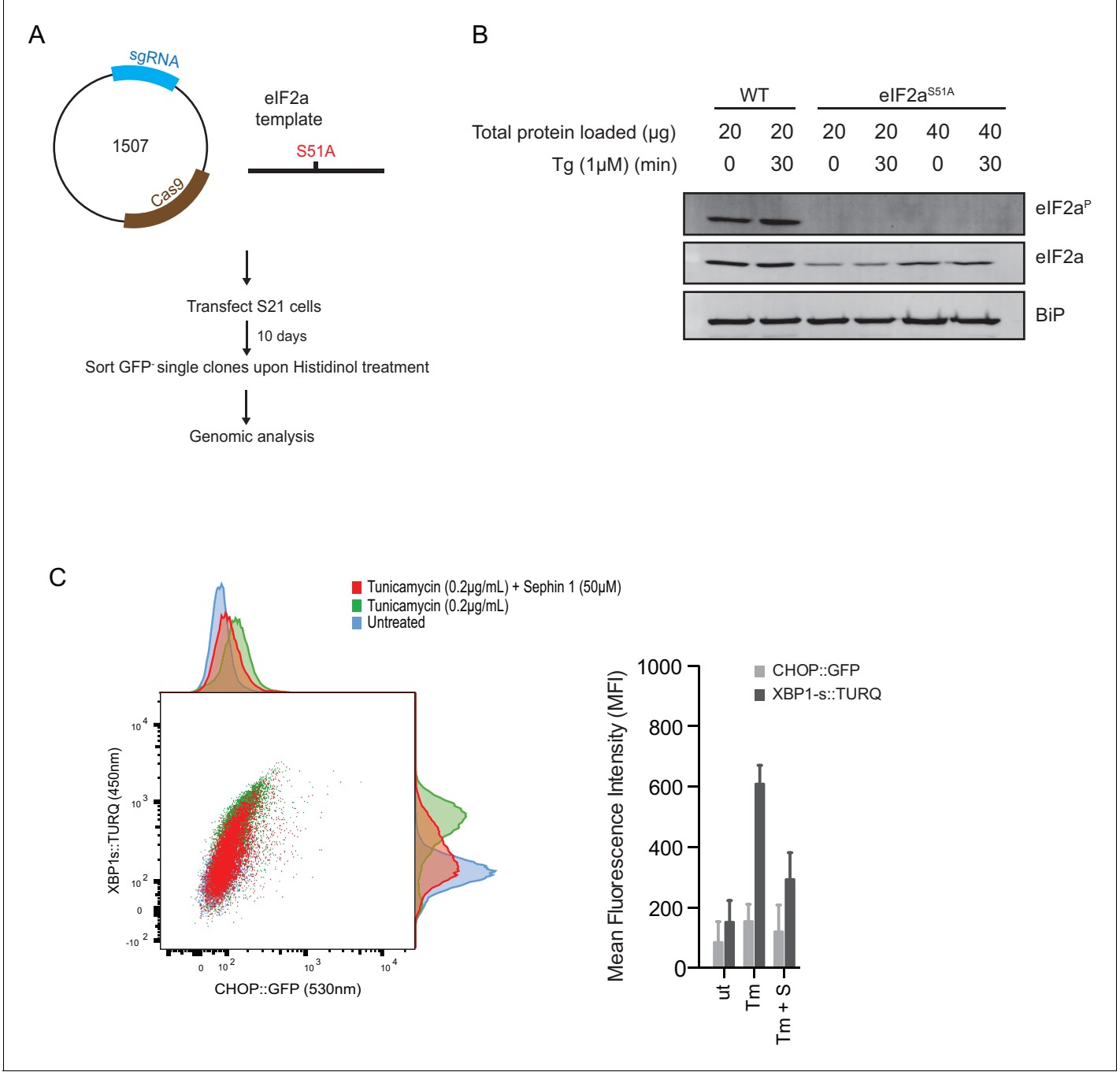

**Figure 12.** ISR-deficient *Eif2s1^{S51A}* (eIF2α^{S51A}) cells retain their responsiveness to Sephin1. (**A**) Schematic representation of procedure used to create dual reporter (CHOP::GFP, XBP1s::Turquoise) *Eif2s1^{S51A}* (eIF2α^{S51A}) CHO-K1 cells using CRISPR-Cas9 system. (**B**) Immunoblot of CHO-K1 cell lysates using anti- eIF2α^P (upper panel), anti- eIF2α (middle panel) and anti-BiP (lower panel) antibodies. Two-fold more cell lysate was loaded onto lanes 5 and 6 to compensate for the lower eIF2α content of the haploid mutant *Eif2s1^{S51A}* cells. (**C**) Two-dimensional plot and histograms of the fluorescent signal of the CHOP::GFP and XBP1s::Turquoise reporters in the *Eif2s1^{S51A}* CHO-K1 cells. Where indicated, the cells were exposed to a low concentration of tunicamycin (0.2 µg/mL; 20 hr) alone or together with Sephin1 (50 µM). The mean ± CV (coefficient of variation) of the fluorescence intensity of the two reporters in each of the two clones is displayed in the bar diagram. Shown is representative experiment of two independent experiments performed. Note the blunted expression of CHOP::GFP wrought by the ISR-defect imposed by the *Eif2s1^{S51A}* mutation.

**Table 1.** Plasmids used.

| Lab number | Lab name | Description | Reference |
|---|---|---|---|
| UK105 | eIF2a-NM_pET30a | His6-tagged mouse eIF2a 1–185 pET-30a(+)" | PMID 15341733 |
| UK168 | PerkKD-pGEX4T-1 | Bacterial expression plasmid for mouse PERK kinase domain | PMID 9930704 |
| UK622 | PGV_PP1G_1–323_V1 | Bacterial expression plasmid forfull-length PP1 phosphatase catalytic domain | PMID 25774600 |
| UK1359 | pSpCas9(BB)—2A-GFP | Mammalian expression of GFP-tagged Cas9 and single guide RNA to introduce double strand breaks (Addgene 48138) | PMID 24157548 |
| UK1367 | pSpCas9(BB)—2A-Puro | Mammalian expression of Puror-tagged Cas9 and single guide RNA to introduce double strand breaks (Addgene 48138) | PMID 24157549 |
| UK1497 | CHO_EIF2K4_guideA*_pSpCas9(BB)—2A-Puro | Puro-tagged CRISPR for targeting human CHO GCN2(EIF2K4) gene | This paper |
| UK1507 | CHO_Eif2s1_guideC_pSpCas9(BB)—2A-Puro | A single guide gRNA plasmid for eIF2a (EIF2S1) locus | This paper |
| UK1599 | CHO_PPP1R15A_guide1_pSpCas9(BB)—2A-GFP | A single guide gRNA plasmid for GADD34 locus | This paper |
| UK1600 | CHO_PPP1R15A_guide2_pSpCas9(BB)—2A-GFP | A single guide gRNA plasmid for GADD34 locus | This paper |
| UK1610 | pSpCas9(BB)—2A-mCherry_V2 | modified pSpCas9(BB)—2A vector to express mCherry together with guide RNA and Cas9 | This paper |
| UK1645 | GST_Myd116_273–657_malE_pGEX_TEV | Bacterial expression of GST-mouse GADD34 273–657 -MBP | This paper |
| UK1677 | huPPP1R15A_325_636_malE_pGEX_TEV | Bacterial expression of GST-human GADD34- MBP | This paper |
| UK1881 | EcBirA_WT_pGEX_TEV (MP1) | Bacterial expression of fastidious E. coli BirA biotin ligase (R118 intact) | This paper |
| UK1897 | mPP1G_1–323_pGEX_TEV_AviTag (MP2) | Bacterial expression GST_TEV_AviTag_FL mPP1G with non-templaled C-term LE | This paper |
| UK1920 | huPPP1R15A_533_624_malE_pGEX_TEV_AviTag (MP1) | Bacterial-expression plasmid for N-tern AviTagged human GADD34 533–624 | This paper |
| UK1921 | huPPP1R15A_325_636_malE_pGEX_TEV_AviTag (MP4) | Bacterial-expression plasmid for N-tern AviTagged human GADD34 325–624 | This paper |
| UK1992 | huPPP1R15A_I596A_533_624_malE_pGEX_TEV_AviTag | Bacterial-expression plasmid for N-term AviTagged human GADD34 533–624, I596A mutation | This paper |
| UK1993 | huPPP1R15A_V556E_R578A_533_624_malE_pGEX_TEV_AviTag | Bacterial-expression plasmid for N-term AviTagged human GADD34 533–624, v556e R578A mutation | This paper |
| UK1994 | huPPP1R15A_V556E_533_624_malE_pGEX_TEV_AviTag | Bacterial-expression plasmid for N-term AviTagged human GADD34 533–624, v556e mutation | This paper |
| UK1995 | huPPP1R15A_F592A_533_624_malE_pGEX_TEV_AviTag | Bacterial-expression plasmid for N-term AviTagged human GADD34 533–624, F592A mutation | This paper |
| UK2081 | CHO_PPP1R15B_guideA_pSpCas9(BB)—2A-mCherry_V2 | guide targeting cgPPP1R15B (hamster CReP) gene 5' end of exon1 | This paper |
| UK2082 | CHO_PPP1R15B_guideB_pSpCas9(BB)—2A-mCherry_V2 | guide targeting cgPPP1R15B (hamster CReP) gene 3' end of exon 1 | This paper |

## Cell treatment, immunoprecipitation and immunoblot

**Antibodies used:** rabbit anti-PPP1R15A (10449–1-AP, ProteinTech, RRID: AB_2168724), rabbit anti-eIF2α-P (ab32157, Abcam, RRID: AB_732117), chicken anti-BiP (*Avezov et al., 2013*), mouse anti-eIF2α (*Scorsone et al., 1987*)

**Drugs used:** tunicamycin (T2250, Melford), thapsigargin (586005, Calbiochem), L-Histidinol (228830010, Acros Organics), PERKi (Gift from GSK, GSK2606414A) Sephin1 (EN300-195090, Enamine)

CHO-K1 cells were plated in 10 cm dishes until they reached 80% confluency, at which point they were treated with 2.5 μg/mL of tunicamycin or DMSO (vehicle) for 7 hr. Cells were washed twice with ice-cold PBS, scraped in presence of PBS with 1 mM EDTA and centrifuged at 376 g for 5 min at 4°C (5424 R, Eppendorf). Four pellet-volume of harvest buffer (10 mM HEPES pH 7.9, 50 mM

NaCl, 0.1 mM EDTA, 0.5% Triton, 0.5 M Sucrose, 1 mM DTT, 4 µg/mL Aprotinin, 1 mM PMSF, 2 µg/mL Pepstatin, 17.5 mM $\beta$-Glycerophosphate, 10 mM Tetrasodium Pyrophosphate, 100 mM NaF) was used to lyse the cells. After 5–10 min of incubation on ice, samples were clarified at 21130 g for 15 min at 4°C (5424 R, Eppendorf). Protein quantification of the clarified supernatants was performed using Bradford method.

For immunoprecipitation, 15 µL Protein A-Sepharose beads (Zymed, 10–1042) per sample where preincubated with anti-PPP1R15A antibody. Equal amounts of protein extract (800 µg) were incubated with the beads over night rotating at 4°C. After four washes with 1 mL of TBS, 20 µL of 2X Laemmli loading buffer were added to the samples. Once incubated at 70°C, same volumes were loaded into a 10% SDS-PAGE gel and transferred to a PVDF membrane.

### Kinase shut-off experiment to assess eIF2α-P dephosphorylation in vivo

The experimental procedure was adapted from (*Chambers et al., 2015*). Briefly, CHO cells (*Gcn2*[-/-]; *Ppp1r15b*[-/-]) were plated in 10 cm dishes at 40% confluency. Sixteen-twenty hours later, fresh media was added and cells were incubated for 2 hr. Sephin1 (50 µM) or DMSO was added to the media for either 30 min or 5 hr before application of thapsigargin (300 nM for 30 min) or tunicamycin (2.5 µg/mL for 2 hr) to induce stress by activation of PERK kinase. GSK2606414A [2 µM] was added to inhibit PERK. The PP1R15A-PP1-dependent decay of the eIF2α-P signal (by its dephosphorylation) was tracked by stopping the reaction at different time points by addition of ice-cold PBS. eIF2α-P and total eIF2α were detected by immunoblot. ImageJ (NIH) was used to quantify signal intensity and one phase decay model was used (GraphPad-Prism V7, RRID: SCR_002798) to analyse the rate decay of eIF2αP dephosphorylation.

### LC-UV-MS analysis

A Shimadzu UFLCXR system coupled to an Applied Biosystems API2000 mass spectrometer was used. Column: Phenomenex Gemini-NX, 3 µm,110 Å, $C_{18}$, 50 × 2 mm (at 40°C). Mobile phase: solvent A: 0.1% formic acid in water; solvent B: 0.1% formic acid in acetonitrile. Gradient: pre-equilibration for 1 min at 5% solvent B in solvent A; then linear gradient 5–98% solvent B over 2 min, 98% B for 2 min, 98–5% B over 0.5 min, then 5% B for 1 min. Flow rate: 0.5 mL/min. Detector: UV detection at 254 nm (channel 1), 220 nm (channel 2). Mass spectrometer: positive ion mode.

## Acknowledgements

We thank S Sharp (MRC epidemiology unit, University of Cambridge) for advice on statistical analysis of the FACS data. R Schulte and the CIMR flow cytometry team for assistance, H P Harding, C Rato-Da Silva, S Preissler, H Sharpe and E Avezov (CIMR) for advice and comments on the manuscript and the Huntington lab (CIMR) for sharing their BLI machine and T Adams (CIMR) for advice on data interpretation. Supported by grants from the Wellcome Trust (Wellcome 200848/Z/16/Z and a strategic award Wellcome 100140). DR is a Wellcome Trust Principal Research Fellow.

## Additional information

#### Competing interests

DR: Member of Elife's Board of Reviewing Editors. The other authors declare that no competing interests exist.

#### Funding

| Funder | Grant reference number | Author |
| --- | --- | --- |
| Wellcome | 200848/Z/16/Z | David Ron |
| Wellcome | 100140 | David Ron |

The funders had no role in study design, data collection and interpretation, or the decision to submit the work for publication.

## Author contributions

AC-C, Led the study, designed and performed the bulk of the experiments and co-wrote the manuscript; JEC, Devised and performed the assay to measure eIF2α-P dephosphorylation in vivo, conceived its utility in evaluating Sephin1 and contributed to writing and editing the manuscript.; PMF, Analysed Sephin1 samples and contributed to the writing and editing of the manuscript; SJM, Contributed to the design and implementation of the assay to measure eIF2α-P dephosphorylation in vivo and contributed to the writing and editing the manuscript; DR, Co-led the study and participated in experimental design, produced some of the expression constructs used, contributed to the interpretation of the data and co-wrote the paper

## Author ORCIDs

Ana Crespillo-Casado, http://orcid.org/0000-0002-7230-3188
Joseph E Chambers, http://orcid.org/0000-0003-4675-0053
Peter M Fischer, http://orcid.org/0000-0002-5866-9271
Stefan J Marciniak, http://orcid.org/0000-0001-8472-7183
David Ron, http://orcid.org/0000-0002-3014-5636

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
