## [Decision Letter]

Thank you for submitting your article "PPP1R15A-mediated eIF2α-P dephosphorylation is unaffected by [(*o*-chlorobenzylidene)amino]guanidines" for consideration by *eLife*. Your article has been reviewed by two peer reviewers, and the evaluation has been overseen by Vivek Malhotra as the Reviewing and Senior Editor. The reviewers have opted to remain anonymous.

The reviews are in agreement on the overall importance and the strength of your data. However, they have requested additional data to strengthen your arguments and to rule out the possibility that differences presented are because of the quality and types of approaches. We would like you to address the following concerns.

1) The most direct comparison of assay type in your paper with those in Das et al. and Tsaytler et al. is the analysis of eIF2α, and even that is done in a different way – thapsigargin versus tunicamycin in the other studies. We realise that you use tunicamycin in the FACS assays, but why not perform a "western blot" experiment that duplicates as much as possible as in the previous published work.

2) Another option would be to repeat the "pull-down" type of experiment done in Figure 3 of Tsaytler et al.

3) Although not related to testing the effects of sephin 1, it would be good to know the effects of salubrinal – the drug that you have identified in your assays.

We hope you will be able to address these issues in a timely manner and submit a stronger revised manuscript. Please do not hesitate to contact us for clarifications and issues with the commentary and handling of your manuscript.

---

## [Author Response]

The reviews are in agreement on the overall importance and the strength of your data. However, they have requested additional data to strengthen your arguments and to rule out the possibility that differences presented are because of the quality and types of approaches. We would like you to address the following concerns.

1) The most direct comparison of assay type in your paper with those in Das et al. and Tsaytler et al. is the analysis of eIF2α, and even that is done in a different way – thapsigargin versus tunicamycin in the other studies. We realise that you use tunicamycin in the FACS assays, but why not perform a "western blot" experiment that duplicates as much as possible as in the previous published work.

2) Another option would be to repeat the "pull-down" type of experiment done in Figure 3 of Tsaytler et al.

In new Figure 10—figure supplement 1 we have applied the kinase shut-off experiment (a method to quantify PPP1R15A dependent eIF2α -P dephosphorylation in vivo) to tunicamycin-treated cells. eIF2α -P dephosphorylation proceeds rapidly in tunicamycin-treated cells. However, Sephin1 had no inhibitory effect on the rate of dephosphorylation. This experiment reveals that under conditions in which Sephin1 exerts its proteostasis-promoting activities, it does not affect rates of eIF2α-P dephosphorylation.

3) Although not related to testing the effects of sephin 1, it would be good to know the effects of salubrinal – the drug that you have identified in your assays.

New Figure 8 addresses this issue. Salubrinal, added at 12 µM (higher concentrations led to conspicuous precipitation) had a mild but reproducible inhibitory effect on in vitro dephosphorylation reactions (inhibition = 22% ± 2.045, unpaired t test, p < 0.0001, n=6). The weakness of salubrinal’s inhibitory effect and the compound’s tendency to precipitate at higher concentrations in the assay buffer frustrated our efforts to establish if inhibition was specific to the eIF2α-P directed ternary complex. Nonetheless, these observations showcase the sensitivity of our assay to even weak inhibitors and strengthen the conclusion regarding Sephin1’s inertness in the same assay. We thank the reviewers for this suggestion.